# A Win-win Deal: Towards Sparse and Robust Pre-trained Language Models

**Yuanxin Liu[1,2,3],∗ Fandong Meng[5], Zheng Lin[1,4]† Jiangnan Li[1,4], Peng Fu[1], Yanan Cao[1,4], Weiping Wang[1], Jie Zhou[5]**
[1]Institute of Information Engineering, Chinese Academy of Sciences
[2]MOE Key Laboratory of Computational Linguistics, Peking University
[3]School of Computer Science, Peking University
[4]School of Cyber Security, University of Chinese Academy of Sciences
[5]Pattern Recognition Center, WeChat AI, Tencent Inc, China
liuyuanxin@stu.pku.edu.cn, {fandongmeng,withtomzhou}@tencent.com
{linzheng,lijiangnan,fupeng,caoyanan,wangweiping}@iie.ac.cn

## Abstract

Despite the remarkable success of pre-trained language models (PLMs), they still face two challenges: First, large-scale PLMs are inefficient in terms of memory footprint and computation. Second, on the downstream tasks, PLMs tend to rely on the dataset bias and struggle to generalize to out-of-distribution (OOD) data. In response to the efficiency problem, recent studies show that dense PLMs can be replaced with sparse subnetworks without hurting the performance. Such subnetworks can be found in three scenarios: 1) the fine-tuned PLMs, 2) the raw PLMs and then fine-tuned in isolation, and even inside 3) PLMs without any parameter fine-tuning. However, these results are only obtained in the in-distribution (ID) setting. In this paper, we extend the study on PLMs subnetworks to the OOD setting, investigating whether sparsity and robustness to dataset bias can be achieved simultaneously. To this end, we conduct extensive experiments with the pre-trained BERT model on three natural language understanding (NLU) tasks. Our results demonstrate that **sparse and robust subnetworks (SRNets) can consistently be found in BERT**, across the aforementioned three scenarios, using different training and compression methods. Furthermore, we explore the upper bound of SRNets using the OOD information and show that **there exist sparse and almost unbiased BERT subnetworks**. Finally, we present 1) an analytical study that provides insights on how to promote the efficiency of SRNets searching process and 2) a solution to improve subnetworks' performance at high sparsity. The code is available at https://github.com/llyx97/sparse-and-robust-PLM.

## 1 Introduction

Pre-trained language models (PLMs) have enjoyed impressive success in natural language processing (NLP) tasks. However, they still face two major problems. On the one hand, the prohibitive model size of PLMs leads to poor efficiency in terms of memory footprint and computational cost [11, 37]. On the other hand, despite being pre-trained on large-scale corpus, PLMs still tend to rely on *dataset bias* [16, 30, 50, 35], i.e., the spurious features of input examples that strongly correlate with the

---

∗Work was done when Yuanxin Liu was a graduate student of IIE, CAS.
†Corresponding author: Zheng Lin.
  Joint work with Pattern Recognition Center, WeChat AI, Tencent Inc, China.

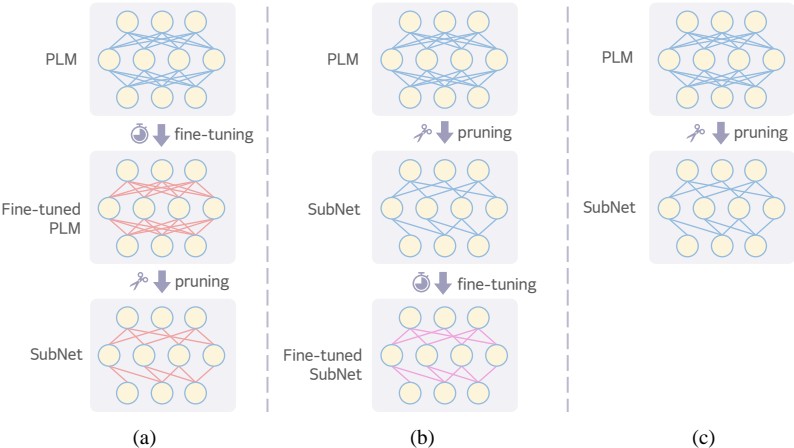

Figure 1: Three kinds of PLM subnetworks obtained from different pruning and fine-tuning paradigms. (a) Pruning a fine-tuned PLM. (b) Pruning the PLM and then fine-tuning the subnetwork. (c) Pruning the PLM without fine-tuning model parameters. The obtained subnetworks are used for testing.

label, during downstream fine-tuning. These two problems pose great challenge to the real-world deployment of PLMs, and they have triggered two separate lines of works.

In terms of the efficiency problem, some recent studies resort to sparse subnetworks as alternatives to the dense PLMs. [23, 31, 25] compress the fine-tuned PLMs in a post-hoc fashion. [3, 32, 27, 24] extend the *Lottery Ticket Hypothesis* (LTH) [8] to search PLMs subnetworks that can be fine-tuned in isolation. Taking one step further, [51] propose to learn task-specific subnetwork structures via mask training [19, 28], without fine-tuning any pre-trained parameter. Fig. 1 illustrates these three paradigms. Encouragingly, the empirical evidences suggest that PLMs can indeed be replaced with sparse subnetworks without compromising the in-distribution (ID) performance.

To address the dataset bias problem, numerous debiasing methods have been proposed. A prevailing category of debiasing methods [4, 41, 21, 18, 35, 12, 42] adjust the importance of training examples, in terms of training loss, according to their bias degree, so as to reduce the impact of biased examples (examples that can be correctly classified based on the spurious features). As a result, the model is forced to rely less on the dataset bias during training and generalizes better to OOD situations.

Although progress has been made in both directions, most existing work tackle the two problems independently. To facilitate real-world application of PLMs, the problems of robustness and efficiency should be addressed simultaneously. Motivated by this, we extend the study on PLM subnetwork to the OOD scenario, investigating **whether there exist PLM subnetworks that are both sparse and robust against dataset bias?** To answer this question, we conduct large-scale experiments with the pre-trained BERT model [5] on three natural language understanding (NLU) tasks that are widely-studied in the question of dataset bias. We consider a variety of setups including the three pruning and fine-tuning paradigms, standard and debiasing training objectives, different model pruning methods, and different variants of PLMs from the BERT family. Our results show that **BERT does contain sparse and robust subnetworks (SRNets)** within certain sparsity constraint (e.g., less than 70%), giving affirmative answer to the above question. Compared with a standard fine-tuned BERT, SRNets exhibit comparable ID performance and remarkable OOD improvement. When it comes to BERT model fine-tuned with debiasing method, SRNets can preserve the full model's ID and OOD performance with much fewer parameters. On this basis, we further explore the upper bound of SRNets by making use of the OOD information, which reveals that **there exist sparse and almost unbiased subnetworks, even in a standard fine-tuned BERT that is biased**.

Regardless of the intriguing properties of SRNets, we find that the subnetwork searching process still have room for improvement, based on some observations from the above experiments. First, we study the timing to start searching SRNets during full BERT fine-tuning, and find that the entire training and searching cost can be reduced from this perspective. Second, we refine the mask training method with gradual sparsity increase, which is quite effective in identifying SRNets at high sparsity.

Our main contributions are summarized as follows:

- We extend the study on PLMs subnetworks to the OOD scenario. To our knowledge, this paper presents the first systematic study on sparsity and dataset bias robustness for PLMs.

- We conduct extensive experiments to demonstrate the existence of sparse and robust BERT subnetworks, across different pruning and fine-tuning setups. By using the OOD information, we further reveal that there exist sparse and almost unbiased BERT subnetworks.

- We present analytical studies and solutions that can help further refine the SRNets searching process in terms of efficiency and the performance of subnetworks at high sparsity.

## 2 Related Work

### 2.1 BERT Compression

Studies on BERT compression can be divided into two classes. The first one focuses on the design of model compression techniques, which include pruning [13, 31, 10], knowledge distillation [34, 38, 20, 26], parameter sharing [22], quantization [47, 49], and combining multiple techniques [39, 29, 25]. The second one, which is based on the lottery ticket hypothesis [8], investigates the compressibility of BERT on different phases of the pre-training and fine-tuning paradigm. It has been shown that BERT can be pruned to a sparse subnetwork after [10] and before fine-tuning [3, 32, 24, 27, 13], without hurting the accuracy. Moreover, [51] show that directly learning subnetwork structures on the pre-trained weights can match fine-tuning the full BERT. In this paper, we follow the second branch of works, and extend the evaluation of BERT subnetworks to the OOD scenario.

### 2.2 Dataset Bias in NLP Tasks

To facilitate the development of NLP systems that truly learn the intended task solution, instead of relying on dataset bias, many efforts have been made recently. On the one hand, challenging OOD test sets are constructed [16, 30, 50, 35, 1] by eliminating the spurious correlations in the training sets, in order to establish more strict evaluation. On the other hand, numerous debiasing methods [4, 41, 21, 18, 35, 12, 42] are proposed to discourage the model from learning dataset bias during training. However, few attention has been paid to the influence of pruning on the OOD generalization ability of PLMs. This work presents a systematic study on this question.

### 2.3 Model Compression and Robustness

Some pioneer attempts have also been made to obtain models that are both compact and robust to adversarial attacks [14, 46, 36, 9, 45] and spurious correlations [48, 7]. Specially, [45, 7] study the compression and robustness question on PLM. Different from [45], which is based on adversarial robustness, we focus on the spurious correlations, which is more common than the worst-case adversarial attack. Compared with [7], which focus on post-hoc pruning of the standard fine-tuned BERT, we thoroughly investigate different fine-tuning methods (standard and debiasing) and subnetworks obtained from the three pruning and fine-tuning paradigms. A more detailed discussion of the relation and difference between our work and previous studies on model compression and robustness is provided in Appendix D.

## 3 Preliminaries

### 3.1 BERT Architecture and Subnetworks

BERT is composed of an embedding layer, a stack of Transformer layers [43] and a task-specific classifier. Each Transformer layer has a multi-head self-attention (MHAtt) module and a feed-forward network (FFN). MHAtt has four kinds of weight matrices, i.e., the query, key and value matrices $\mathbf{W}_{Q,K,V} \in \mathbb{R}^{d_{\text{model}} \times d_{\text{model}}}$, and the output matrix $\mathbf{W}_{AO} \in \mathbb{R}^{d_{\text{model}} \times d_{\text{model}}}$. FFN consits of two linear layers $\mathbf{W}_{\text{in}} \in \mathbb{R}^{d_{\text{model}} \times d_{\text{FFN}}}$, $\mathbf{W}_{\text{out}} \in \mathbb{R}^{d_{\text{FFN}} \times d_{\text{model}}}$, where $d_{\text{FFN}}$ is the hidden dimension of FFN.

To obtain the subnetwork of a model $f(\boldsymbol{\theta})$ parameterized by $\boldsymbol{\theta}$, we apply a binary pruning mask $\mathbf{m} \in \{0, 1\}^{|\boldsymbol{\theta}|}$ to its weight matrices, which produces $f(\mathbf{m} \odot \boldsymbol{\theta})$, where $\odot$ is the Hadamard product.

For BERT, we focus on the $L$ Transformer layers and the classifier. The parameters to be pruned are $\boldsymbol{\theta}_{pr} = \{\mathbf{W}_{\text{cls}}\} \cup \{\mathbf{W}_Q^l, \mathbf{W}_K^l, \mathbf{W}_V^l, \mathbf{W}_{AO}^l, \mathbf{W}_{\text{in}}^l, \mathbf{W}_{\text{out}}^l\}_{l=1}^L$, where $\mathbf{W}_{\text{cls}}$ is the classifier weights.

## 3.2 Pruning Methods

### 3.2.1 Magnitude-based Pruning

Magnitude-based pruning [17, 8] zeros-out parameters with low absolute values. It is usually realized in an iterative manner, namely, iterative magnitude pruning (IMP). IMP alternates between pruning and training and gradually increases the sparsity of subnetworks. Specifically, a typical IMP algorithm consists of four steps: (i) Training the full model to convergence. (ii) Pruning a fraction of parameters with the smallest magnitude. (iii) Re-training the pruned subnetwork. (iv) Repeat (ii)-(iii) until reaching the target sparsity. To obtain subnetworks from the pre-trained BERT, i.e., (b) and (c) in Fig. 1, the subnetwork parameters are rewound to the pre-trained values after (iii), and (i) can be abandoned. More details about our IMP implementations can be found in Appendix A.1.1.

### 3.2.2 Mask Training

Mask training treats the pruning mask $\mathbf{m}$ as trainable parameters. Following [28, 51, 33, 27], we achieve this through binarization in forward pass and gradient estimation in backward pass.

Each weight matrix $\mathbf{W} \in \mathbb{R}^{d_1 \times d_2}$, which is frozen during mask training, is associated with a bianry mask $\mathbf{m} \in \{0, 1\}^{d_1 \times d_2}$, and a real-valued mask $\hat{\mathbf{m}} \in \mathbb{R}^{d_1 \times d_2}$. In the forward pass, $\mathbf{W}$ is replaced with $\mathbf{m} \odot \mathbf{W}$, where $\mathbf{m}$ is derived from $\hat{\mathbf{m}}$ through binarization:

$$\mathbf{m}_{i,j} = \begin{cases} 1 & \text{if } \hat{\mathbf{m}}_{i,j} \geq \phi \\ 0 & \text{otherwise} \end{cases} \tag{1}$$

where $\phi$ is the threshold. In the backward pass, since the binarization operation is not differentiable, we use the *straight-through estimator* [2] to compute the gradients for $\hat{\mathbf{m}}$ using the gradients of $\mathbf{m}$, i.e., $\frac{\partial \mathcal{L}}{\partial \mathbf{m}}$, where $\mathcal{L}$ is the loss. Then, $\hat{\mathbf{m}}$ is updated as $\hat{\mathbf{m}} \leftarrow \hat{\mathbf{m}} - \eta \frac{\partial \mathcal{L}}{\partial \mathbf{m}}$, where $\eta$ is the learning rate.

Following [33, 27], we initialize the real-valued masks according to the magnitude of the original weights. The complete mask training algorithm is summarized in Appendix A.1.2.

## 3.3 Debiasing Methods

As described in the Introduction, the debiasing methods measure the bias degree of training examples. This is achieved by training a *bias model*. The inputs to the bias model are hand-crafted spurious features based on our prior knowledge of the dataset bias (Section 4.1.3 describes the details). In this way, the bias model mainly relies on the spurious features to make predictions, which can then serve as a measurement of the bias degree. Specifically, given the bias model prediction $\mathbf{p}_b = (\mathbf{p}_b^1, \cdots, \mathbf{p}_b^K)$ over the $K$ classes, the bias degree $\beta = \mathbf{p}_b^c$, i.e., the the probability of the ground-truth class $c$.

Then, $\beta$ can be used to adjust the training loss in several ways, including *product-of-experts* (PoE) [4, 18, 21], *example reweighting* [35, 12] and *confidence regularization* [41]. Here we describe the standard cross-entropy and PoE, and the other two methods are introduced in Appendix A.2.

**Standard Cross-Entropy** computes the cross-entropy between the predicted distribution $\mathbf{p}_m$ and the ground-truth one-hot distribution $\mathbf{y}$ as $\mathcal{L}_{\text{std}} = -\mathbf{y} \cdot \log \mathbf{p}_m$.

**Product-of-Experts** combines the predictions of main model and bias model, i.e., $\mathbf{p}_b$ and $\mathbf{p}_m$, and then computes the training loss as $\mathcal{L}_{\text{poe}} = -\mathbf{y} \cdot \log \text{softmax} (\log \mathbf{p}_m + \log \mathbf{p}_b)$.

## 3.4 Notations

Here we define some notations, which will be used in the following sections.

- $\mathcal{A}_{\mathcal{L}}^t(f(\boldsymbol{\theta}))$: Training $f(\boldsymbol{\theta})$ with loss $\mathcal{L}$ for $t$ steps, where $t$ can be omitted for simplicity.
- $\mathcal{P}_{\mathcal{L}}^p(f(\boldsymbol{\theta}))$: Pruning $f(\boldsymbol{\theta})$ using pruning method $p$ and training loss $\mathcal{L}$.
- $\mathcal{M}(f(\mathbf{m}\boldsymbol{\theta}))$: Extracting the pruning mask of $f(\mathbf{m}\boldsymbol{\theta})$, i.e., $\mathcal{M}(f(\mathbf{m}\boldsymbol{\theta})) = \mathbf{m}$.

- $\mathcal{L} \in \{\mathcal{L}_{\text{std}}, \mathcal{L}_{\text{poe}}, \mathcal{L}_{\text{reweight}}, \mathcal{L}_{\text{confreg}}\}$ and $p \in \{\text{imp}, \text{imp-rw}, \text{mask}\}$, where "imp" and "imp-rw"denote the standard IMP and IMP with weight rewinding, as described in Section 3.2.1. "mask" stands for mask training.
- $\mathcal{E}_d(f(\boldsymbol{\theta}))$: Evaluating $f(\boldsymbol{\theta})$ on the test data with distribution $d \in \{\text{ID}, \text{OOD}\}$.

# 4 Sparse and Robust BERT Subnetworks

## 4.1 Experimental Setups

### 4.1.1 Datasets and Evaluation

**Natural Language Inference** We use MNLI [44] as the ID dataset for NLI. MNLI is comprised of premise-hypothesis pairs, whose relationship may be *entailment*, *contradiction*, or *neutral*. In MNLI the word overlap between premise and hypothesis is strongly correlated with the *entailment* class. To solve this problem, the OOD HANS dataset [30] is built so that such correlation does not hold.

**Paraphrase Identification** The ID dataset for paraphrase identification is QQP [3], which contains question pairs that are labelled as either *duplicate* or *non-duplicate*. In QQP, high lexical overlap is also strongly associated with the *duplicate* class. The OOD datasets PAWS-qqp and PAWS-wiki [50] are built from sentences in Quora and Wikipedia respectively. In PAWS sentence pairs with high word overlap have a balanced distribution over *duplicate* and *non-duplicate*.

**Fact Verification** FEVER [4] [40] is adopted as the ID dataset of fact verification, where the task is to assess whether a given evidence *supports* or *refutes* the claim, or whether there is *not-enough-info* to reach a conclusion. The OOD dataset Fever-Symmetric (v1 and v2) [35] is proposed to evaluate the influence of the claim-only bias (the label can be predicted correctly without the evidence).

For NLI and fact verification, we use Accuracy as the evaluation metric. For paraphrase identification, we evaluate using the F1 score. More details of datasets and evaluation are shown in Appendix B.1.

### 4.1.2 PLM Backbone

We mainly experiment with the BERT-base-uncased model [5]. It has roughly 110M parameters in total, and 84M parameters in the Transformer layers. As described in Section 3.1, we derive the subnetworks from the Transformer layers and report sparsity levels relative to the 84M parameters. To generalize our conclusions to other PLMs, we also consider two variants of the BERT family, namely RoBERTa-base and BERT-large, the results of which can be found in Appendix C.5.

### 4.1.3 Training Details

Following [4], we use a simple linear classifier as the bias model. For HANS and PAWS, the spurious features are based on the the word overlapping information between the two input text sequences. For Fever-Symmetric, the spurious features are max-pooled word embeddings of the claim sentence. More details about the bias model and the spurious features are presented in Appendix B.3.1.

Mask training and IMP basically use the same hyper-parameters (adopting from [42]) as full BERT. An exception is longer training, because we find that good subnetworks at high sparsity levels require more training to be found. Unless otherwise specified, we select the best checkpoints based on the ID dev performance, without using OOD information. All the reported results are averaged over 4 runs. We defer training details about each dataset, and each training and pruning setup, to Appendix B.3.

## 4.2 Subnetworks from Fine-tuned BERT

### 4.2.1 Problem Formulation and Experimental Setups

Given the fine-tuned full BERT $f(\boldsymbol{\theta}_{ft}) = \mathcal{A}_{\mathcal{L}_1}(f(\boldsymbol{\theta}_{pt}))$, where $\boldsymbol{\theta}_{pt}$ and $\boldsymbol{\theta}_{ft}$ are the pre-trained and fine-tuned parameters respectively, the goal is to find a subnetwork $f(\mathbf{m} \odot \boldsymbol{\theta}'_{ft}) = \mathcal{P}^p_{\mathcal{L}_2}(f(\boldsymbol{\theta}_{ft}))$ that

---

[3] https://www.kaggle.com/c/quora-question-pairs
[4] See the licence information at https://fever.ai/download/fever/license.html

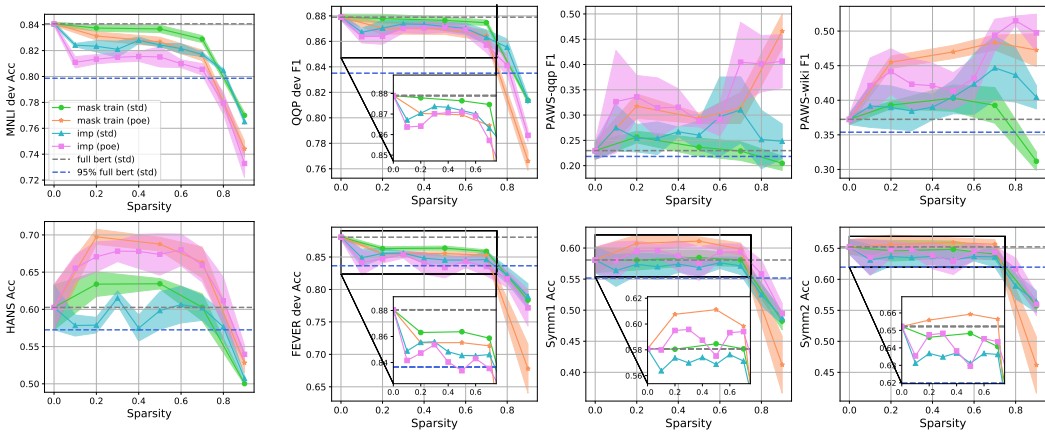

Figure 2: Results of subnetworks pruned from the CE fine-tuned BERT. "std" means standard, and the shadowed areas denote standard deviations, which also apply to the other figures of this paper.

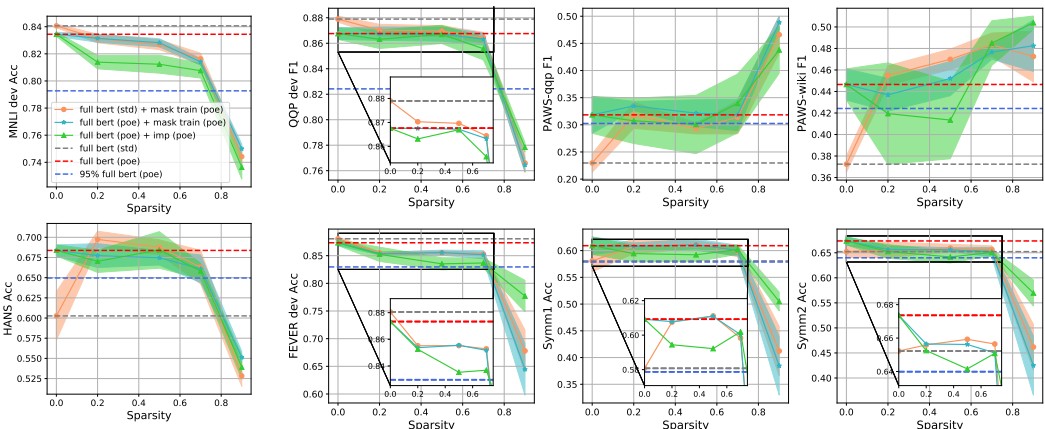

Figure 3: Results of subnetworks pruned from the PoE fine-tuned BERT. Results of the "mask train (poe)" subnetworks from Fig. 2 (the orange line) are also reported for reference.

satisfies a target sparsity level $s$ and maximize the ID and OOD performance.

$$\max_{\mathbf{m}, \boldsymbol{\theta}'_{ft}} \left( \mathcal{E}_{\mathrm{ID}} \left( f \left( \mathbf{m} \odot \boldsymbol{\theta}'_{ft} \right) \right) + \mathcal{E}_{\mathrm{OOD}} \left( f \left( \mathbf{m} \odot \boldsymbol{\theta}'_{ft} \right) \right) \right), \text{ s.t. } \frac{\|\mathbf{m}\|_0}{|\boldsymbol{\theta}_{pr}|} = (1-s) \qquad (2)$$

where $\|\|_0$ is the $L_0$ norm and $|\boldsymbol{\theta}_{pr}|$ is the total number of parameters to be pruned. In practice, the above optimization problem is achieved via $\mathcal{P}^p_{\mathcal{L}_2}()$, which minimizes the loss $\mathcal{L}_2$ on the ID training set. When the pruning method is IMP, the subnetwork parameters will be further fine-tuned and $\boldsymbol{\theta}'_{ft} \neq \boldsymbol{\theta}_{ft}$. For mask training, only the subnetwork structure is updated and $\boldsymbol{\theta}'_{ft} = \boldsymbol{\theta}_{ft}$.

We consider two kinds of fine-tuned full BERT, which utilize the standard CE loss and PoE loss respectively (i.e., $\mathcal{L}_1 \in \{\mathcal{L}_{\mathrm{std}}, \mathcal{L}_{\mathrm{poe}}\}$). IMP and mask training are used as the pruning methods (i.e., $p \in \{\mathrm{imp}, \mathrm{mask}\}$). For the standard fine-tuned BERT, both $\mathcal{L}_{\mathrm{std}}$ and $\mathcal{L}_{\mathrm{poe}}$ are examined in the pruning process. For the PoE fine-tuned BERT, we only use $\mathcal{L}_{\mathrm{poe}}$ during pruning. Note that in this work, we mainly experiment with $\mathcal{L}_{\mathrm{std}}$ and $\mathcal{L}_{\mathrm{poe}}$. $\mathcal{L}_{\mathrm{reweight}}$ and $\mathcal{L}_{\mathrm{confreg}}$ are also examined for subnetworks from fine-tuned BERT, the results of which can be found in Appendix C.1.

### 4.2.2 Results

**Subnetworks from Standard Fine-tuned BERT** The results are shown in Fig. 2 (In this paper, we present most results in figures for clear comparisons. Actual values of the results can be found in the code link.). We discuss them from three perspectives. For the full BERT, we can see that standard

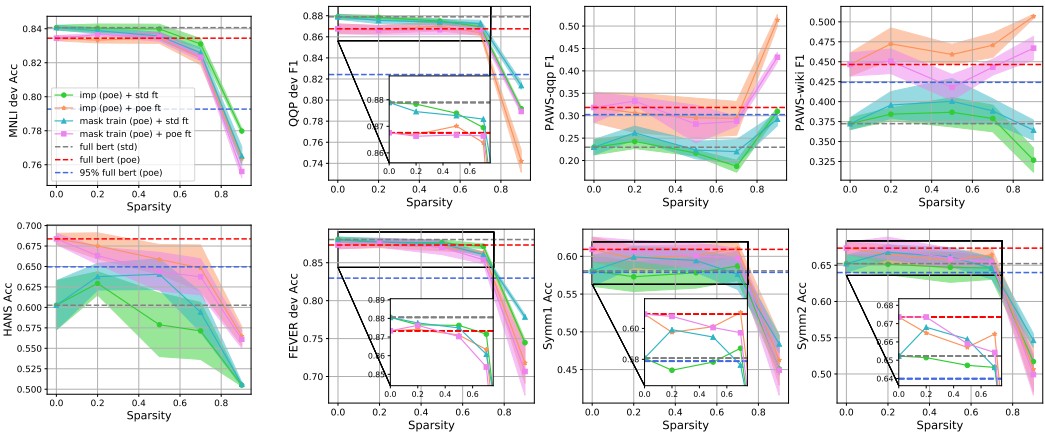

Figure 4: Results of BERT subnetworks fine-tuned in isolation. "ft" is short for fine-tuning.

CE fine-tuning, which achieves good results on the ID dev sets, performs significantly worse on the OOD test sets. This demonstrates that the ID performance of BERT depends, to a large extent, on memorizing the dataset bias.

In terms of the subnetworks, we can derive the following observations: (1) Using any of the four pruning methods, we can compress a large proportion of the BERT parameters (up to 70% sparsity) and still preserve 95% of the full model's ID performance. (2) With standard pruning, i.e., "mask train (std)" or "imp (std)", we can observe small but perceivable improvement over the full BERT on the HANS and PAWS datasets. This suggests that pruning may remove some parameters related to the bias features. (3) The OOD performance of "mask train (poe)" and "imp (poe)" subnetworks is even better, and the ID performance degrades slightly but is still above 95% of the full BERT. This shows that introducing the debiasing objective in the pruning process is beneficial. Specially, as mask training does not change the model parameters, the results of "mask train (poe)" implicates that the biased "full bert (std)" contains sparse and robust subnetworks (SRNets) that already encode a less biased solution to the task. (4) SRNets can be identified across a wide range of sparsity levels (from 20% ∼ 70%). However at higher sparsity of 90%, the performance of the subnetworks is not desirable. (5) We also find that there is an abnormal increase of the PAWS F1 score at 70% ∼ 90% sparsity for some pruning methods, when the corresponding ID performance drops sharply. This is because the class distribution of PAWS is imbalanced (see Appendix B.1), and thus even a naive random-guessing model can outperform the biased full model on PAWS. Therefore, the OOD improvement should only be acceptable when there is no large ID performance decline.

Comparing IMP and mask training, the latter performs better in general, except for "mask train (poe)" at 90% sparsity on QQP and FEVER. This suggests that directly optimizing the subnetwork structure is a better choice than using the magnitude heuristic as the pruning metric.

**Subnetworks from PoE Fine-tuned BERT**    Fig. 3 presents the results. We can find that: (1) For the full BERT, the OOD performance is obviously promoted with the PoE debiasing method, while the ID performance is sacrificed slightly. (2) Unlike the subnetworks from the standard fine-tuned BERT, the subnetworks of PoE fine-tuned BERT (the green and blue lines) cannot outperform the full model. However, these subnetworks maintain comparable performance at up to 70% sparsity, on both the ID and OOD settings, making them desirable alternatives to the full model in resource-constraint scenarios. Moreover, this phenomenon suggests that there is a great redundancy of BERT parameters, even when OOD generalization is taken into account. (3) With PoE-based pruning, subnetworks from the standard fine-tuned BERT (the orange line) is comparable with subnetworks from the PoE fine-tuned BERT (the blue line). This means we do not have to fine-tune a debiased BERT before searching for the SRNets. (4) IMP, again, slightly underperforms mask training at moderate sparsity levels, while it is better at 90% sparsity on the fact verification task.

### 4.3 BERT Subnetworks Fine-tuned in Isolation

#### 4.3.1 Problem Formulation and Experimental Setups

Given the pre-trained BERT $f(\boldsymbol{\theta}_{pt})$, a subnetwork $f(\mathbf{m} \odot \boldsymbol{\theta}_{pt})$ is obtained before downstream fine-tuning. The goal is to maximize the performance of the fine-tuned subnetwork $\mathcal{A}_{\mathcal{L}_1}(f(\mathbf{m} \odot \boldsymbol{\theta}_{pt}))$:

$$\max_{\mathbf{m}} \left( \mathcal{E}_{\mathrm{ID}} \left( \mathcal{A}_{\mathcal{L}_1}(f(\mathbf{m} \odot \boldsymbol{\theta}_{pt})) \right) + \mathcal{E}_{\mathrm{OOD}} \left( \mathcal{A}_{\mathcal{L}_1}(f(\mathbf{m} \odot \boldsymbol{\theta}_{pt})) \right) \right), \text{ s.t. } \frac{\|\mathbf{m}\|_0}{|\boldsymbol{\theta}_{pr}|} = (1 - s) \quad (3)$$

Following the LTH [8], we solve this problem using the train-prune-rewind pipeline. For IMP, the procedure is described in Section 3.2.1 and $\mathbf{m} = \mathcal{M}(\mathcal{P}_{\mathcal{L}_2}^{\mathrm{imp\text{-}rw}}(f(\boldsymbol{\theta}_{pt})))$. For mask training, the subnetwork structure is learned from $f(\boldsymbol{\theta}_{ft})$ (same as the previous section) and $\mathbf{m} = \mathcal{M}(\mathcal{P}_{\mathcal{L}_2}^{\mathrm{mask}}(f(\boldsymbol{\theta}_{ft})))$.

We employ CE and PoE loss for model fine-tuning (i.e., $\mathcal{L}_1 \in \{\mathcal{L}_{\mathrm{std}}, \mathcal{L}_{\mathrm{poe}}\}$). Since we have shown that using the debiasing loss in pruning is conducive, the CE loss is not considered (i.e., $\mathcal{L}_2 = \mathcal{L}_{\mathrm{poe}}$).

#### 4.3.2 Results

The results of subnetworks fine-tuned in isolation are presented in Fig. 4. It can be found that: (1) For standard CE fine-tuning, the "mask train (poe)" subnetworks are superior to "full bert (std)" on the OOD test data, i.e., the subnetworks are less susceptible to the dataset bias during training. (2) In terms of the PoE-based fine-tuning, the "imp (poe)" and "mask train (poe)" subnetworks are generally comparable to "full bert (poe)". (3) For most of the subnetworks, "poe ft" clearly outperforms "std ft" in the OOD setting, which suggests that it is important to use the debiasing method in fine-tuning, even if the BERT subnetwork structure has already encoded some unbiased information.

Moreover, based on (1) and (2), we can extend the LTH on BERT [3, 32, 24, 27]: **The pre-trained BERT contains SRNets that can be fine-tuned in isolation, using either standard or debiasing method, and match or even outperform the full model in both the ID and OOD evaluations.**

### 4.4 BERT Subnetworks Without Fine-tuning

#### 4.4.1 Problem Formulation and Experimental Setups

This setup aims at finding a subnetwork $f(\mathbf{m} \odot \boldsymbol{\theta}_{pt})$ inside the pre-trained BERT, which can be directly employed to a task. The problem is formulated as:

$$\max_{\mathbf{m}} \left( \mathcal{E}_{\mathrm{ID}} \left( f(\mathbf{m} \odot \boldsymbol{\theta}_{pt}) \right) + \mathcal{E}_{\mathrm{OOD}} \left( f(\mathbf{m} \odot \boldsymbol{\theta}_{pt}) \right) \right), \text{ s.t. } \frac{\|\mathbf{m}\|_0}{|\boldsymbol{\theta}_{pr}|} = (1 - s) \quad (4)$$

Following [51], we fix the pre-trained parameters $\boldsymbol{\theta}_{pt}$ and optimize the mask variables $\mathbf{m}$. This process can be represented as $\mathcal{P}_{\mathcal{L}}^{\mathrm{mask}}(f(\boldsymbol{\theta}_{pt}))$, where $\mathcal{L} \in \{\mathcal{L}_{\mathrm{std}}, \mathcal{L}_{\mathrm{poe}}\}$.

#### 4.4.2 Results

As we can see in Fig. 5: (1) With CE-based mask training, the identified subnetworks (under $50\%$ sparsity) in pre-trained BERT are competitive with the CE fine-tuned full BERT. (2) Similarly, using PoE-based mask training, the subnetworks under $50\%$ sparsity are comparable to the PoE fine-tuned full BERT, which demonstrates that SRNets for a particular downstream task already exist in the pre-trained BERT. (3) "mask train (poe)" subnetworks in pre-trained BERT can even match the subnetworks found in the fine-tuned BERT (the orange lines) in some cases (e.g., on PAWS and on FEVER under $50\%$ sparsity). Nonetheless, the latter exhibits a better overall performance.

### 4.5 Sparse and Unbiased BERT Subnetworks

#### 4.5.1 Problem Formulation and Experimental Setups

To explore the upper bound of BERT subnetworks in terms of OOD generalization, we include the OOD training data in mask training, and use the OOD test sets for evaluation. Like the previous sections, we investigate three pruning and fine-tuning paradigms, as formulated by Eq. 2, 3 and 4 respectively. We only consider the standard CE for subnetwork and full BERT fine-tuning, which is more vulnerable to the dataset bias. Appendix B.3.3 summarizes the detailed experimental setups.

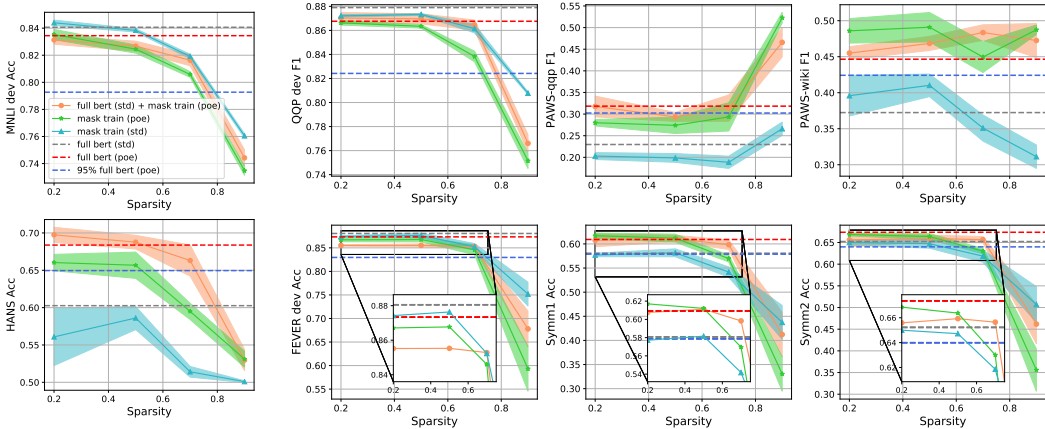

Figure 5: Results of BERT subnetworks without fine-tuning. Results of the "mask train (poe)" subnetworks from Fig. 2 (the orange line) are also reported for reference.

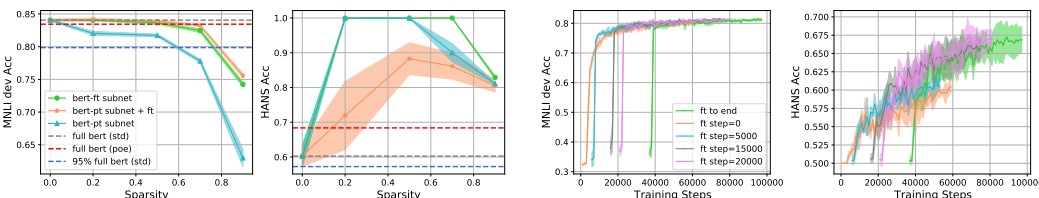

Figure 6: NLI results of BERT subnetworks found using the OOD information. Results of the other two tasks can be found in Appendix C.2.

Figure 7: NLI mask training curves (70% sparse), starting from BERT fine-tuned for varied steps. Appendix C.3 shows results of the other two tasks.

### 4.5.2 Results

From Fig. 6 we can observe that: (1) The subnetworks from fine-tuned BERT ("bert-ft subnet") at $20\% \sim 70\%$ sparsity achieve nearly $100\%$ accuracy on HANS, and their ID performance is also close to the full BERT. (2) The subnetworks in the pre-trained BERT ("bert-pt subnet") also have very high OOD accuracy, while they perform worse than "bert-ft subnet" in the ID setting. (3) "bert-pt subnet + ft" subnetworks, which are fine-tuned in isolation with CE loss, exhibits the best ID performance, and the poorest OOD performance. However, compared to the full BERT, these subnetworks still rely much less on the dataset bias, reaching nearly $90\%$ HANS accuracy at $50\%$ sparsity. Jointly, these results show that there consistently exist BERT subnetworks that are almost unbiased towards the MNLI training set bias, under the three kinds of pruning and fine-tuning paradigms.

## 5    Refining the SRNets Searching Process

In this section, we study how to further improve the SRNets searching process based on mask training, which generally performs better than IMP, as shown in Section 4.2 and Section 4.3.

### 5.1    The Timing to Start Searching SRNets

Compared with searching subnetworks from the fine-tuned BERT, directly searching from the pre-trained BERT is more efficient in that it dispenses with fine-tuning the full model. However, the former has a better overall performance, as we have shown in Section 4.4. This induces a question: **At which point of the BERT fine-tuning process, can we find subnetworks comparable to those found after the end of fine-tuning using mask training?** To answer this question, we perform mask training on the model checkpoints $f(\boldsymbol{\theta}_t) = \mathcal{A}^t_{\mathcal{L}_{\text{std}}}(f(\boldsymbol{\theta}_{pt}))$ from different steps $t$ of BERT fine-tuning.

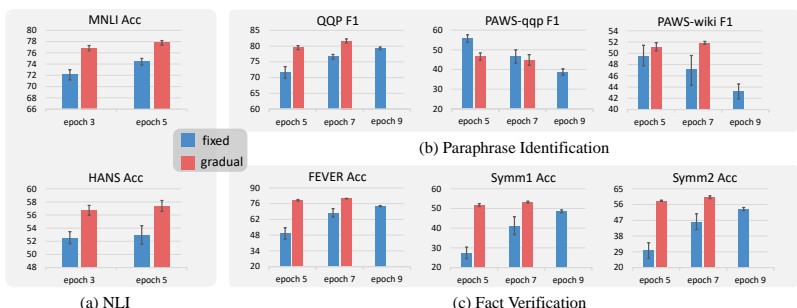

Figure 8: Comparison between fixed sparsity and gradual sparsity increase for mask training with the standard fine-tuned full BERT. The subnetworks are at 90% sparsity.

Fig. 7 shows the mask training curves, which start from different $f(\boldsymbol{\theta}_t)$. We can see that "ft step=0" converges slower and to a worse final accuracy, as compared with "ft to end", especially on the HANS dataset. However, with 20,000 steps of full BERT fine-tuning, which is roughly 55% of the "ft to end", the mask training performance is very competitive. This suggests that the total training cost of SRNet searching can be reduced, by a large amount, in the full model training stage.

To actually reduce the training cost, we need to predict the exact timing to start mask training. This is intractable without information of all the training curves in Fig. 7. A feasible solution is adopting the idea of early-stopping (see Appendix E.1 for detailed discussions). However, accurately predicting the optimal timing (with the least amount of fine-tuning and comparable subnetwork performance to fully fine-tuning) is indeed difficult and we invite follow-up studies to investigate this question.

### 5.2 SRNets at High Sparsity

As the results of Section 4 demonstrate, there is a sharp decline of the subnetworks' performance from $70\% \sim 90\%$ sparsity. We conjecture that this is because directly initializing mask training to $90\%$ reduces the model's capacity too drastically, and thus causes some difficulties in optimization. Therefore, we gradually increase the sparsity from $70\% \sim 90\%$ during mask training, using the cubic sparsity schedule [52] (see Appendix C.4 for ablation studies). Fig. 8 compares the fixed sparsity used in the previous sections and the gradual sparsity increase, across varied mask training epochs. We find that while simply extending the training process is conducive, gradual sparsity increase achieves better results. In particular, "gradual" outperforms "fixed" with lower training cost on all the three tasks, except for the PAWS dataset, A similar phenomenon is explained in Section 4.2.2.

## 6 Conclusions and Limitations

In this paper, we investigate whether sparsity and robustness to dataset bias can be achieved simultaneously for PLM subnetworks. Through extensive experiments, we demonstrate that BERT indeed contains sparse and robust subnetworks (SRNets) across a variety of NLU tasks and training and pruning setups. We further use the OOD information to reveal that there exist sparse and almost unbiased BERT subnetworks. Finally, we present analysis and solutions to refine the SRNet searching process in terms of subnetwork performance and searching efficiency.

The limitations of this work is twofold. First, we focus on BERT-like PLMs and NLU tasks, while dataset biases are also common in other scenarios. For example, gender and racial biases exist in dialogue generation systems [6] and PLMs [15]. In the future work, we would like to extend our exploration to other types of PLMs and NLP tasks (see Appendix E.2 for a discussion). Second, as we discussed in Section 5.1, our analysis on "the timing to start searching SRNets" mainly serves as a proof-of-concept, and actually reducing the training cost requires predicting the exact timing.

## Acknowledgments and Disclosure of Funding

This work was supported by National Natural Science Foundation of China (61976207 and 61906187).

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
