# A Win-win Deal: Towards Sparse and Robust Pre-trained Language Models

**Yuanxin Liu[1,2,3],[\*] Fandong Meng[5], Zheng Lin[1,4],[†] Jiangnan Li[1,4], Peng Fu[1], Yanan Cao[1,4],**
**Weiping Wang[1], Jie Zhou[5]**
[1]Institute of Information Engineering, Chinese Academy of Sciences
[2]MOE Key Laboratory of Computational Linguistics, Peking University
[3]School of Computer Science, Peking University
[4]School of Cyber Security, University of Chinese Academy of Sciences
[5]Pattern Recognition Center, WeChat AI, Tencent Inc, China
liuyuanxin@stu.pku.edu.cn, {fandongmeng,withtomzhou}@tencent.com
{linzheng,lijiangnan,fupeng,caoyanan,wangweiping}@iie.ac.cn

## A  More Information of Pruning and Debiasing Methods

### A.1  Pruning Methods

#### A.1.1  Iterative Magnitude Pruning

Algo. 1 summarizes our implementation of IMP and IMP with weight rewinding. In practice, we set the per time pruning ratio $\Delta s = 10\%$ and the pruning interval $\Delta t = 0.1 \cdot t_{\max}$.

#### A.1.2  Mask Training

As we described in Section 3.2.2 of the main paper, we realize mask training via binarization in forward pass and gradient estimation in backward pass. Following [17, 11], we adopt a magnitude-based strategy to initialize the real-valued masks. Specially, we consider two variants: The first one (hard variant) identifies the weights in matrix $\mathbf{W}$ with the smallest magnitudes, and sets the corresponding elements in $\hat{\mathbf{m}}$ to zero, and the remaining elements to a fixed value:

$$\hat{\mathbf{m}}_{i,j} = \begin{cases} 0 & \text{if } \mathbf{W}_{i,j} \in \text{Min}_s(\text{abs}(\mathbf{W})) \\ \alpha \times \phi & \text{otherwise} \end{cases} \tag{1}$$

where $\text{Min}_s(\text{abs}(\mathbf{W}))$ extracts the weights with the lowest absolute value, according to sparsity level $s$. $\alpha \geq 1$ is a hyper-parameter. The second one (soft variant) directly utilizes the absolute values of the weights for mask initialization:

$$\hat{\mathbf{m}}_{i,j} = \text{abs}(\mathbf{W}_{i,j}) \tag{2}$$

To control the sparsity of the model, the threshold $\phi$ is adjusted dynamically at a frequency of $\Delta t_\phi$ training steps. In practice, we control the sparsity in a local way, i.e., all the weight matrices $\mathbf{W} \in \boldsymbol{\theta}_{pr}$ should satisfy the same sparsity constraint $s$. Algo. 2 summarizes the entire process of mask training.

### A.2  Debiasing Methods

We have introduced the PoE method in Section 3.3. Here we provide descriptions of the other two debiasing methods, i.e., example reweighting and confidence regularization.

---

[\*]Work was done when Yuanxin Liu was a graduate student of IIE, CAS.
[†]Corresponding author: Zheng Lin.
 Joint work with Pattern Recognition Center, WeChat AI, Tencent Inc, China.

36th Conference on Neural Information Processing Systems (NeurIPS 2022).

**Algorithm 1:** Iterative Magnitude Pruning (+ weight rewinding)

---

**Input:** PLM $f(\boldsymbol{\theta}_0)$ w. $\boldsymbol{\theta}_0 = \boldsymbol{\theta}_{ft}$, maximum training steps $t_{\max}$, pruning interval $\Delta t$, per time pruning ratio $\Delta s$, target sparsity level $s = k \cdot \Delta s$ ($k \in \{1, 2, \cdots\}$), pruning method $p \in \{\text{imp}, \text{imp-rw}\}$

**Output:** Pruned subnetwork $f(\mathbf{m} \odot \boldsymbol{\theta}'_{ft})$

**1** Initialize the pruning mask $\mathbf{m} = 1^{|\boldsymbol{\theta}_0|}$ and the number of pruning $n = 0$
**2** **while** $t < t_{max}$ **do**
**3**    **if** *(t mod $\Delta t$) == 0* **then**
**4**      `# For imp, return the subnetwork after some further training`
**5**      **if** $n \cdot \Delta s == s$ *and* $p ==$imp **then**
**6**        **return** $f(\mathbf{m} \odot \boldsymbol{\theta}_t)$
**7**      **end**
**8**      Prune $\Delta s \cdot |\boldsymbol{\theta}_0|$ from the remaining parameters $\mathbf{m} \odot \boldsymbol{\theta}_t$ based on the magnitudes, and update $\mathbf{m}$ accordingly
**9**      $n \leftarrow n + 1$
**10**      `# For imp-rw, return the subnetwork directly after pruning`
**11**      **if** $n \cdot \Delta s == s$ *and* $p ==$imp-rw **then**
**12**        **return** $f(\mathbf{m} \odot \boldsymbol{\theta}_0)$
**13**      **end**
**14**    **end**
**15**    Update the remaining model parameters $\mathbf{m} \odot \boldsymbol{\theta}_t$ via AdamW [12];
**16** **end**

---

**Example Reweighting** directly assigns an importance weight to the standard CE training loss, according to the bias degree $\beta$:

$$\mathcal{L}_{\text{reweight}} = -(1 - \beta)\,\mathbf{y} \cdot \log \mathbf{p}_m \tag{3}$$

**Confidence Regularization** is based on knowledge distillation [9]. It involves a teacher model trained with the standard CE loss. The teacher model's prediction $\mathbf{p}_t$ is used as a supervision signal to train the main model. To account for the bias degree of training examples, $\mathbf{p}_t$ is smoothed using a scaling function $\text{S}(\mathbf{p}_t, \beta)$, and the final loss is computed as:

$$\mathcal{L}_{\text{confreg}} = -\text{S}(\mathbf{p}_t, \beta) \cdot \log \mathbf{p}_m$$
$$\text{S}(\mathbf{p}_t, \beta) = \frac{(\mathbf{p}_t^j)^{(1-\beta)}}{\sum_{k=1}^{K}(\mathbf{p}_t^k)^{(1-\beta)}} \tag{4}$$

# B  More Experimental Setups

## B.1  Datasets and Evaluations

We utilize eight datasets from three NLU tasks. The statistics of different dataset splits are summarized in Tab. 1. If one dataset has a test set, we use it for evaluation, and otherwise we report results on the dev set. For MNLI and QQP, since the official test server [3] only allows two submissions a day, we instead evaluate on the dev sets, following [2, 11, 19]. For FEVER, we use the training and evaluation data processed by [20] [4].

Tab. 2 shows the distribution of examples over classes. We can see that the distributions of the QQP and PAWS$_{qqp}$ evaluation sets are imbalanced. Specially, in the OOD PAWS$_{qqp}$, where a biased model tends to predict most examples to the *duplicate* class, simply classifying all examples as *non-duplicate* can achieve substantial improvement in accuracy (from $28.2\%$ to $71.8\%$). To account for this, we use the F1 score to evaluate the performance on the three paraphrase identification datasets. Specifically, we calculate the weighted average of the F1 score of each class. However, the class imbalance may still affect the evaluation on PAWS (as we discussed in Section 4.2.2) and therefore the OOD improvement should be assessed by also considering the ID performance.

---

[3] `https://gluebenchmark.com/`
[4] `https://github.com/TalSchuster/FeverSymmetric`

**Algorithm 2:** Mask Training

**Input:** PLM $f(\boldsymbol{\theta}_0)$ w. $\boldsymbol{\theta}_0 \in \{\boldsymbol{\theta}_{pt}, \boldsymbol{\theta}_{ft}\}$, maximum training steps $t_{\max}$, frequency $\Delta t_\phi$, target sparsity level $s$, threshold $\phi$, hyper-parameter $\alpha$, initialization method $init \in \{\text{hard}, \text{soft}\}$

**Output:** Pruned subentwork $f(\mathbf{m} \odot \boldsymbol{\theta}_0)$

1 **if** $init$ == hard **then**
2      Initialize the real-valued mask $\hat{\mathbf{m}}$ according to Eq. 1
3      Set threshold $\phi = 0.01$
4 **else**
5      Initialize the real-valued mask $\hat{\mathbf{m}}$ according to Eq. 2
6      Set threshold $\phi$ according to the sparsity constraint
7 **end**
8 **while** $t < t_{max}$ **do**
9      Get a mini-batch of $B$ examples $\{(\mathbf{x}_b, y_b)\}_{b=1}^B$
10      Forward pass through binarization:

11          $\mathcal{L}(f(\mathbf{x}_b, \mathbf{m} \odot \boldsymbol{\theta}_0), y_b), \qquad \text{where } \mathbf{m}_{i,j} = \begin{cases} 1 & \text{if } \hat{\mathbf{m}}_{i,j} \geq \phi \\ 0 & \text{otherwise} \end{cases}$

12      Backward pass through gradient estimation:
13          $\hat{\mathbf{m}} \leftarrow \hat{\mathbf{m}} - \eta \frac{\partial \mathcal{L}}{\partial \hat{\mathbf{m}}}$
14      **if** *(t mod $\Delta t_\phi$) == 0* **then**
15          Update the threshold $\phi$ to satisfy the sparsity constraint
16      **end**
17 **end**
18 **return** $f(\mathbf{m} \odot \boldsymbol{\theta}_0)$

Table 1: The number of examples in different dataset splits. The splits used for evaluation are highlighted with red color. The dev set for MNLI is MNLI-m.

| | NLI | | Paraphrase Identification | | | Fact Verification | | |
| --- | --- | --- | --- | --- | --- | --- | --- | --- |
| | MNLI | HANS | QQP | PAWS-qqp | PAWS-wiki | FEVER | FEVER-Symm1 | FEVER-Symm2 |
| Train | 392,702 | 30,000 | 363,849 | 11,988 | 49,401 | 242,911 | - | - |
| Dev | 9,815 | 30,000 | 40,432 | 677 | 8,000 | 16,664 | - | 708 |
| Test | - | - | - | - | 8,000 | - | 717 | 712 |

## B.2 Software and Computational Resources

We use two types of GPU, i.e., Nvidia V100 and TITAN RTX. All the experiments are run on a single GPU. Our codes are based on the Pytorch[5] and the huggingface transformers library[6] [26].

## B.3 Training Details

### B.3.1 Bias Model

As mentioned in Section 4.1.3, we train the bias model with spurious features. For MNLI and QQP, we adopt the hand-crafted word overlapping features proposed by [3], which includes:

- Whether all the hypothesis words also belong to the premise.
- Whether the hypothesis appears as a continuous subsequence in the premise.
- The percentage of the hypothesis words $\mathbf{w}^h = \{\mathbf{w}_1^h, \mathbf{w}_2^h, \cdots, \mathbf{w}_{|\mathbf{w}^h|}^h\}$ that appear in the premise $\mathbf{w}^p = \{\mathbf{w}_1^p, \mathbf{w}_2^p, \cdots, \mathbf{w}_{|\mathbf{w}^p|}^p\}$. Formally $\frac{|\mathbf{w}^h \cap \mathbf{w}^p|}{|\mathbf{w}^h|}$.
- The average of the maximum similarity between each hypothesis word and all the premise words: $\frac{1}{|\mathbf{w}^h|} \text{sum}(\{\max(\{\text{sim}(\mathbf{w}_i^p, \mathbf{w}_j^h) | \forall \mathbf{w}_j^p \in \mathbf{w}^p\}) | \forall \mathbf{w}_i^h \in \mathbf{w}^h\})$, where the similarity is computed based on the fastText word vectors [15] and the cosine distance.

---

[5]https://pytorch.org/
[6]https://github.com/huggingface/transformers

Table 2: Data distribution over classes. The meaning of the abbreviations are: ent (entailment), cont (contradiction), dulp (duplicate), supp (support), not-info (not-enough-info). "Eval" represents the dataset split used for evaluation, as described in Tab. 1

| | | MNLI | HANS |
|---|---|---|---|
| Train | ent | 33.3% | 50% |
| | cont | 33.3% | 50% |
| | neutral | 33.3% | 0% |
| Eval | ent | 35.4% | 50% |
| | cont | 32.7% | 50% |
| | neutral | 31.8% | 0% |

| | | QQP | PAWS$_{qqp}$ | PAWS$_{wiki}$ |
|---|---|---|---|---|
| Train | dulp | 36.9% | 31.5% | 44.2% |
| | non-dulp | 63.1% | 68.5% | 55.8% |
| Eval | dulp | 36.8% | 28.2% | 44.2% |
| | non-dulp | 63.2% | 71.8% | 55.8% |

| | | FEVER | Symm1 | Symm2 |
|---|---|---|---|---|
| Train | supp | 41.4% | - | - |
| | refute | 17.2% | - | - |
| | not-info | 41.4% | - | - |
| Eval | supp | 47.9% | 52.9% | 50% |
| | refute | 52.1% | 47.1% | 50% |
| | not-info | 0% | 0% | 0% |

Table 3: Basic training hyper-parameters.

| | #Epoch | Learning Rate | Batch Size | Max Length | Eval Interval | Eval Metric | Optimizer |
|---|---|---|---|---|---|---|---|
| MNLI | 3 or 5 | 5e-5 | 32 | 128 | 1,000 | Acc | AdamW |
| QQP | 3 | 2e-5 | 32 | 128 | 1,000 | F1 | AdamW |
| FEVER | 3 | 2e-5 | 32 | 128 | 500 | Acc | AdamW |

- The minimum of the same similarities above: $\min(\{\max(\{\text{sim}(\mathbf{w}_i^p, \mathbf{w}_j^h)|\forall \mathbf{w}_j^p \in \mathbf{w}^p\})|\forall \mathbf{w}_i^h \in \mathbf{w}^h\})$.

For FEVER, we use the max-pooled word embeddings of the claim sentence, which are also based on the fastText word vectors.

### B.3.2 Full BERT

The main training hyper-parameters are shown in Tab. 3, which basically follow [25]. Most of the hyper-parameters are the same for different training strategies, except for the number of training epochs (#Epoch) on MNLI. For the standard CE loss and example reweighting, the model is trained for 3 epochs. For PoE and confidence regularization, the model is trained for 5 epochs.

### B.3.3 Mask Training and IMP

Mask training and IMP basically use the same set of hyper-parameters as full BERT, except for longer training. The number of training epochs for mask training and IMP is 5 on MNLI, and 7 on QQP and FEVER. The hyper-parameters specific to mask training or IMP are summarized in Tab. 4. Unless otherwise specified, we adopt the hard-variant of mask initialization (Eq. 1) and fix the subnetwork sparsity to target sparsity $s$ throughout the process of mask training. Some special experimental setups are described as follows:

**Subnetworks from Fine-tuned BERT** When we search for subnetworks at low sparsity (e.g., 20%) from a fine-tuned BERT, we find that mask training (with debiasing loss) stably improves the OOD performance, while the ID performance peaks at an early point of training and then slightly drops and recovers later. Therefore, the ID performance favors the early checkpoints, which are not good at the OOD generalization. To address this problem, we select the best checkpoint after $0.7 \cdot t_{\max}$ of training, but still according to the performance on the ID dev set. This strategy is only adopted for mask training on fine-tuned BERT (for all sparsity levels), and in other cases we select the best checkpoint across training based on ID performance.

**BERT Subnetworks Fine-tuned in Isolation** When fine-tuning the searched subnetworks (with their weights rewound to pre-trained values) in isolation, we use the same set of hyper-parameters as full BERT fine-tuning.

**Sparse and Unbiased BERT Subnetworks** The OOD data is used in this setup. Specifically, we utilize the training data of HANS and PAWS for NLI and paraphrase identification respectively. In terms of the FEVER-Symmetric dataset, which does not provide a training set (see Tab. 1), we use the dev set of FEVER-Symm2 and copy the data 10 times to construct the OOD training data. The OOD and ID training data are then combined to form the final training set. Note that the evaluation sets are the same as the other setups, and **NO** test data is used in mask training.

Table 4: Basic hyper-parameters related to pruning methods. $t_{\max}$ is the number of optimization steps by training #Epoch epochs.

| | Mask Training | | | | IMP | |
|---|---|---|---|---|---|---|
| Mask Init | Sparsity Schedule | $\phi$ | $\alpha$ | $\Delta t_\phi$ | $\Delta s$ | $\Delta t$ |
| magnitude (hard) | fixed to $s$ | 0.01 | 2 | equal to Eval Interval | 10% | $0.1 \cdot t_{\max}$ |

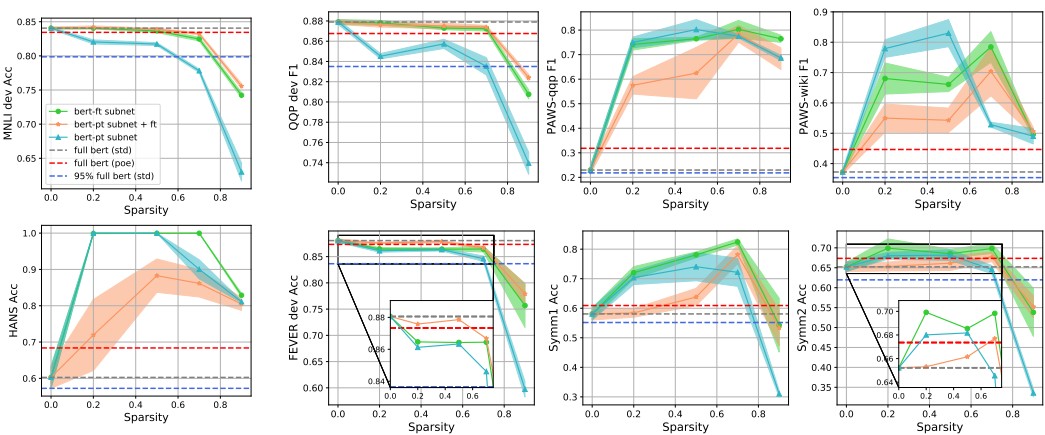

Figure 1: Results of subnetworks pruned from the CE fine-tuned BERT, with different debiasing methods in pruning.

**Gradual Sparsity Increase** We mainly experiment with the gradual sparsity increase schedule for subnetworks at 90% sparsity. Concretely, we increase the sparsity from 70% to 90% during the process of mask training. The real-valued mask is initialized using the soft-variant (Eq. 2). This is because we find that the hard-variant is difficult to optimize with sparsity increase.

## C   More Results and Analysis

### C.1   More Debiasing Methods

In Section 4, we mainly experiment with the PoE debiasing method. Here, we combine mask training with the other two debiasing methods, namely example reweighting and confidence regularization, and search for SRNets from the CE fine-tuned BERT. Fig. 1 presents the results. As we can see: (1)

Figure 2: Results of subnetworks found using the OOD information.

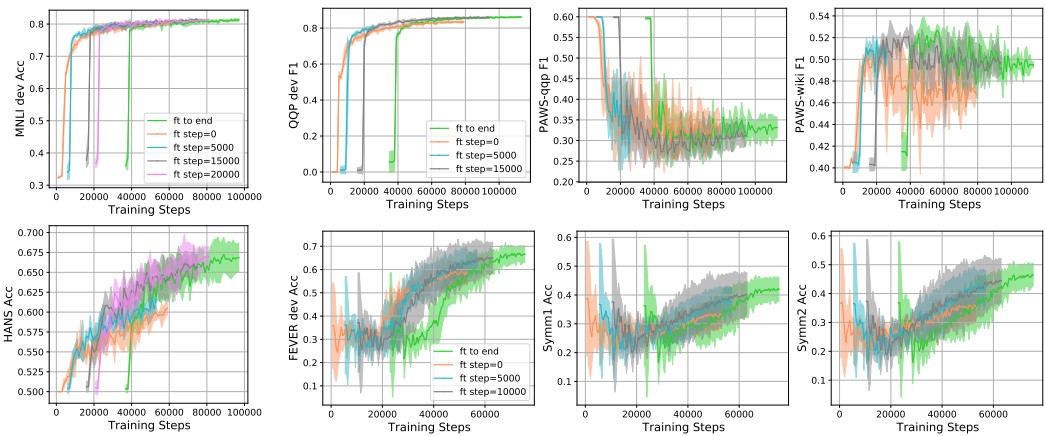

Figure 3: Mask training curves starting from full BERT checkpoints fine-tuned for varied steps. The sparsity levels are 70%, 70% and 90% for MNLI, QQP and FEVER respectively. At these sparsity levels, the gap between "ft step=0" and "ft to end" is the largest, according to Fig. 5 of the main paper.

Pruning with different debiasing methods almost consistently improves the OOD performance over the CE fine-tuned BERT. (2) The confidence regularization method (the grey lines) only achieves mild OOD improvement over the full BERT, while it preserves more ID performance compared with the other two methods. This phenomenon is in accordance with the results from [24], which propose the confidence regularization method to achieve a better trade-off between the ID and OOD performance.

## C.2 Sparse and Unbiased Subnetworks

Fig. 2 shows the results of mask training with the OOD training data. We can see that the general patterns in paraphrase identification and fact verification datasets are basically the same as the NLI datasets. Although the identified subnetworks cannot achieve 100% accuracy on PAWS and FEVER-Symmetric as on HANS, they substantially narrow the gap between OOD and ID performance, as compared with the full BERT. An exception is on the Symm2, where the upper bound of SRNets seems not very high. This is probably because we do not have enough examples (708 in total) to represent the data distribution of the FEVER-Symmetric dataset. Therefore, we conjecture that the existence of sparse and unbiased subnetworks might be ubiquitous.

## C.3 The Timing to Start Searching SRNets

Fig. 3 shows the mask training curves on all the 8 datasets. Similar to the NLI datasets, mask training on the other two tasks can achieve comparable results as "ft to end" by starting from an intermediate checkpoint of BERT fine-tuning. For QQP, we can start from 15,000 steps of full BERT fine-tuning (44% of $t_{max}$). For FEVER, we can start from 10,000 steps (44% of $t_{max}$).

## C.4 Ablation Studies on Gradual Sparsity Increase

As we mentioned in Appendix B.3.3, we increase the sparsity from 70% to 90% and adopt the soft variant of mask initialization. To explain the reason for using this specific strategy, we present the ablation study results in Tab. 5. We can observe that: (1) Replacing the hard variant of mask initialization with the soft variant is beneficial, which leads to obvious improvements on the QQP, FEVER, Symm1 and Symm2 datasets. (2) Gradually increasing the sparsity further promotes the performance, with the 0.7~0.9 strategy achieving the best results on 7 out of the 8 datasets.

## C.5 Results on RoBERTa-base and BERT-large

It has been shown by [8, 23] that pre-trained model RoBERTa [10] have better OOD generalization than BERT. [23] also shows that larger PLMs, which are more computationally expensive, are more

Table 5: Ablation studies of the gradual sparsity increase schedule. The number of training epochs are 3, 5 and 5 for MNLI, QQP and FEVER respectively. The subnetworks are at $90\%$ sparsity. The numbers in the subscripts are standard deviations.

| | | MNLI | HANS | | | QQP | $PAWS_{qqp}$ | $PAWS_{qqp}$ | | | FEVER | Symm1 | Symm2 |
|---|---|---|---|---|---|---|---|---|---|---|---|---|---|
| fixed | hard | $72.09_{0.92}$ | $52.56_{0.92}$ | fixed | hard | $71.64_{1.85}$ | $\mathbf{55.70_{1.92}}$ | $49.59_{1.84}$ | fixed | hard | $49.56_{5.09}$ | $27.45_{2.94}$ | $29.75_{4.40}$ |
| | soft | $72.63_{0.31}$ | $52.82_{0.47}$ | | soft | $77.08_{0.66}$ | $46.48_{3.55}$ | $49.38_{0.98}$ | | soft | $72.80_{0.95}$ | $46.67_{0.73}$ | $52.33_{0.75}$ |
| gradual | 0.2∼0.9 | $73.61_{0.28}$ | $53.90_{0.87}$ | gradual | 0.2∼0.9 | $75.79_{0.39}$ | $51.57_{0.69}$ | $47.94_{0.98}$ | gradual | 0.2∼0.9 | $73.53_{1.36}$ | $46.47_{1.66}$ | $52.42_{1.39}$ |
| | 0.5∼0.9 | $75.06_{0.31}$ | $54.99_{1.28}$ | | 0.5∼0.9 | $77.54_{0.47}$ | $50.92_{0.97}$ | $48.86_{0.89}$ | | 0.5∼0.9 | $77.01_{0.43}$ | $49.87_{0.95}$ | $56.57_{0.22}$ |
| | 0.7∼0.9 | $\mathbf{76.84_{0.46}}$ | $\mathbf{56.72_{0.75}}$ | | 0.7∼0.9 | $\mathbf{79.49_{0.58}}$ | $46.59_{1.81}$ | $\mathbf{51.15_{0.73}}$ | | 0.7∼0.9 | $\mathbf{79.01_{0.68}}$ | $\mathbf{51.74_{0.71}}$ | $\mathbf{58.17_{0.33}}$ |

Table 6: Results of RoBERTa-base and BERT-large on the NLI task. We conduct mask training with PoE loss on the standard fine-tuned PLMs. "0.5∼0.7" denotes gradual sparsity increase. The numbers in the subscripts are standard deviations.

| RoBERTa-base | | MNLI | HANS | BERT-large | | MNLI | HANS |
|---|---|---|---|---|---|---|---|
| full model | std | $87.14_{0.21}$ | $68.33_{0.88}$ | full model | std | $86.84_{0.13}$ | $69.44_{2.39}$ |
| | poe | $86.56_{0.18}$ | $76.15_{1.35}$ | | poe | $86.25_{0.17}$ | $76.27_{1.55}$ |
| mask train | 0.5 | $85.40_{0.14}$ | $75.17_{0.55}$ | mask train | 0.5 | $85.47_{0.28}$ | $75.40_{0.64}$ |
| | 0.7 | $83.48_{0.29}$ | $68.63_{1.33}$ | | 0.7 | $77.54_{6.10}$ | $60.19_{7.56}$ |
| | 0.5∼0.7 | $84.41_{0.15}$ | $71.95_{1.23}$ | | 0.5∼0.7 | $84.83_{0.26}$ | $70.18_{2.24}$ |

robust. To examine whether our conclusions can generalize to RoBERTa and larger versions of BERT, we conduct mask training on the standard fine-tuned RoBERTa-base and BERT-large models and use the PoE debiasing loss in the mask training process.

The results are shown in Tab. 6. We can see that, for RoBERTa-base: (1) At 50% sparsity, the searched subnetworks outperform the full RoBERTa (std) by 6.84 points on HANS, with a relative small drop of 1.74 on MNLI, validating that SRNets can be found in RoBERTa. (2) At 70% sparsity, the vanilla mask training produces subnetworks with undesirable ID performance and OOD performance comparable to full model (std). In comparison, when we gradually increase the sparsity level from 50% to 70%, the ID and OOD performance are improved simultaneously, demonstrating that gradual sparsity increase is also effective for RoBERTa.

When it comes to BERT-large, the conclusions are basically the same as BERT-base and RoBERTa-base: (1) We can find 50% sparse SRNets from BERT-large using the original mask training. (2) Gradual sparsity increase is also effective for BERT-large. Additionally, we find that the original mask training exhibits high variance at 70% sparsity because the training fails for some random seeds. In comparison, with gradual sparsity increase, the searched subnetworks have better performance and low variance.

## D   Related Work on Model Compression and Robustness

Some prior attempts have also been made to obtain compact and robust deep neural networks. We discuss the relationship and difference between these works and our paper from three perspectives:

**Robustness Types**   There are various types of model robustness, including generalization to in-distribution unseen examples, robustness towards dataset bias [1, 14, 31, 20] and adversarial attacks [6], etc. Among the researches on model compression and robustness, adversarial robustness [7, 28, 22, 5, 27] and dataset bias robustness [29, 4] are the most widely studied. In this paper, we focus on the dataset bias problem, which is more common than the worst-case adversarial attack, in terms of real-world application.

**Compression Methods**   A major direction in robust model compression is about the design of compression methods. [21] investigate the effect of magnitude-based pruning on adversarially trained models. [7, 28] treat sparsity and adversarial robustness as a constrained optimization problem, and solve it using the alternating direction method of multipliers (ADMM) framework [30]. [22, 29, 13] combine learnable weight mask (i.e., mask training) and robust training objectives. Our study

investigates the use of magnitude-based pruning and mask training, which are also widely employed in the literature of BERT compression.

**Application Fields**   Despite the topic of model compression and robustness has been proposed for years, it is mostly studied in the context of computer vision (CV) tasks and models, and few attention has been paid to the NLP field. Considering the real-world application potential of PLMs, it is critical to study the questions of PLM compression and robustness jointly. To this end, some recent studies extend the evaluation of compressed PLMs to consider adversarial robustness [27] and dataset bias robustness [4].

Although our work shares the same topic with [4], we differ in several aspects. First, the scope and focus of our research questions are different. They aim at analyzing the impact of different compression methods (pruning and knowledge distillation [9]) on the OOD robustness of standard fine-tuned BERT. By contrast, we focus on subnetworks obtained from different pruning and fine-tuning paradigms and consider both standard fine-tuning and debiasing fine-tuning. Second, our conclusions are different. The results of [4] suggest that pruning generally has a negative impact on the robustness of BERT. In comparison, we revel the consistent existence of sparse BERT subnetworks that are more robust to dataset bias than the full model.

## E   More Discussions

### E.1   How to Predict the Timing to Start Searching SRNets?

A feasible way of solution is to stop full BERT fine-tuning when there is no significant improvement across several consecutive evaluation steps. The patience of early-stopping can be determined based on the computational budget. If our resource is limited, we can at least directly training the mask on $\theta_{pt}$, which can still produce SRNets at 50% sparsity (as shown by Section 4.4.2).

### E.2   How to Generalize to Other Scenarios?

In this work, we focus on NLU tasks and PLMs from the BERT family. However, the methodology we utilize is agnostic to the type of bias, task and backbone model. Theoretically, it can be flexibly adapted to other scenarios by simply change the spurious features to train the bias model (for the three debiasing methods considered in this paper) or combine the pruning method with another kind of debiasing method that also involves model training. In the future work, we would like to extend our exploration to other types of PLMs (e.g., language generation models like GPT [16] and T5 [18]) and other types of NLP tasks (e.g., dialogue generation).