# OpenReview forum: "A Win-win Deal: Towards Sparse and Robust Pre-trained Language Models"
_NeurIPS.cc/2022/Conference — NeurIPS 2022 Accept_

### Official Review · Reviewer_Spmm · 2022-07-11

**Rating:** 6
**Confidence:** 3
**Soundness:** 3 good
**Presentation:** 2 fair
**Contribution:** 3 good

**Summary:**

* The paper presents an analysis of the intersection of sparsification methods and robustness vis-a-vis in-domain (ID) and out-of-domain (OOD) performance, showing that sparse and robust networks (dubbed SRNets in the paper) can be found for pre-trained BERT models, finetuned BERT models and sparsified pre-trained models that are then finetuned in isolation.
* Concretely, the paper considers 2 pruning methods: Iterative Magnitude Pruning and Mask Training; and 2 training paradigms: standard finetuning and Product of Experts (PoE), the latter to improve OOD performance. They present their results on 3 datasets: MNLI (with HANS as OOD), QQP (with PAWS-wiki and PAWS-qqp as OOD datasets) and FEVER (with FEVER Symmetric v1 and v2 as OOD datasets).
* For pruned finetuned models, the authors show that the sparsified models achieve competitive performance compared to full model performance upto 70% sparsity, and achieve better than finetuned OOD performance. When trained with PoE, the sparsified models achieve competitive ID and OOD performance at 70% sparsity.
* The authors show that for sparsified pre-trained models, finetuned in isolation, the models tend to achieve better OOD performance, showing that SRNets can be found in pre-trained BERT models, and can be fine-tuned in isolation.
* Further the authors show that for BERT subnetworks without finetuning, the pruned networks are competitive with full finetuned models both while using standard and PoE objectives.
* Finally, the authors show that even using intermediate finetuned checkpoints for pruning can be used to achieve good finetuning performance, and also present a methodology to achieve improved performance for high sparsity networks.

**Questions:**

Questions for the Authors
1.  Was there any reason why PoE variant (as opposed to the Debiasing Focal Loss variant) was considered for the experiments ?
2. I am not sure what line 63 means ("Second, we ameliorate the mask training method ..."). Would it be possible to rephrase it ?
3.  For the results on BERT Subnetworks Fine-tuned in Isolation, it seems like IMP (PoE) + std ft. does considerably worse compared to full bert std finetuning on the OOD dataset (HANS) compared to Mask train (PoE) + std ft. I am curious if there is any hypothesis on why this might be the case ?
4. The IMP methods seem to have a considerably higher variance compared to the Mask Train methods. Is there a hypothesis on why this might be the case ?

Typographic Edits:
1. Line 8: even inside 3) PLM without -> 3) even inside PLM without
2. Line 49: BERT do contain -> BERT does contain

**Limitations:**

The authors have considered the limitations fairly well. One minor addition that might be helpful would be to elaborate a bit more on how would the exploration on other type of tasks look like (eg: what sort of biases would you want to investigate w.r.t language generation and how would you adapt the proposed methodology for the same).

**Strengths And Weaknesses:**

Strengths:
1. The paper presents a comprehensive analysis on a novel area in the intersection of OOD robustness and sparsification of finetuned models, which is quite relevant from a usability standpoint
2. The existence of SPNets in the different setups considered is quite intriguing, and does present an interesting direction of work.
3. The results of being able to combine PoE and pruning methods is quite useful from a practical standpoint for deploying models in the wild.

Weaknesses:
1. My main concern with is with Section 4.1. The authors claim that they "refine the SRNet searching process to improve efficiency .." (Line 69). However, from Section 4.1, while the authors show that starting from earlier steps can yield competitive results compared to full finetuning, the authors make no recommendations on how to find such a step a-priori (eg: as a function of the model / dataset etc). For example, why should the 20k steps be what is considered, and not any other step-count. Or is it possible to predict at what step count should we start the pruning process, without actually finetuning the entire model. Overall, given that this is listed as a main contribution of the work, it would have been nice to have a more in-depth analysis of this topic.
2. While the results presented as figures does help analyse trends, it would be helpful to also include the actual values in a Table as well. Otherwise, it makes comparisons (especially across the different setups) extremely hard (eg: between BERT subnetworks without Finetuning vs BERT subnetworks on finetuned models vs BERT subnetworks finetuned in isolation).
3. It would be good to consider at least one additional pre-trained model to show some generality of observations across models (Eg: RoBERTa[1], especially since RoBERTa has been shown to have better OOD generalization than BERT [2,3] ).
4. Additionally, OOD robustness is observed more for large models compared to base models ([2]). Thus it would be good to see how much value add would the proposed methods bring for those scenarios. Specifically, would it still be the case that pruning yields comparable to better OOD performance, or would the loss in model capacity result in worse OOD performance compared to the baseline methods.
5. [Minor] Line 125 mentions example reweighing and confidence regularization also being considered for the work, but it is not mentioned otherwise in the paper. It would be good to include those results as well.


References

[1] Liu, Yinhan, et al. "Roberta: A robustly optimized bert pretraining approach." arXiv preprint arXiv:1907.11692 (2019).

[2] Tu, Lifu, et al. "An empirical study on robustness to spurious correlations using pre-trained language models." Transactions of the Association for Computational Linguistics 8 (2020): 621-633.

[3] Hendrycks, Dan, et al. "Pretrained transformers improve out-of-distribution robustness." arXiv preprint arXiv:2004.06100 (2020).

---

> ### Author Response · Authors · 2022-08-02
> **Response to Reviewer Spmm (Part 2/2)**
>
> ### Q6: For BERT subnetworks fine-tuned in isolation, why does IMP (PoE) + std ft perform worse than Mask train (PoE) + std ft?
> According to our results, mask training outperforms IMP in general. We conjecture that the reason is two-fold: First, mask training directly optimizes the subnetwork structure towards the given objective, which maybe more effective than IMP, which heuristically retains the weights with larger absolute values. Second, in IMP, once a weight is pruned, it is discarded permanently. By contrast, mask training allows pruned connections to re-grow in the entire training process, which is more flexible.
>
> ### Q7: Why do IMP methods seem to have a higher variance than mask training?
> Interesting question. We have also observed this phenomenon. However, we think the current results are insufficient to draw any reasonable hypothesis.
>
> ### Q8: Elaborate on how to explore other types of tasks
> Thanks for the good suggestion, we will revise the limitation part to include more detailed discussions. In addition to NLU tasks, biases are also common generation tasks. For example, social biases (e.g., gender or racial) exist in dialogue generation systems [1] and PLMs [2].
>
> Theoretically, our approach can be flexibly adapted to other types of biases or tasks. We can simply change the spurious features to train the bias model (for the three debiasing methods considered in this paper) or combine the pruning method with another kind of debiasing method that also involves model training.
>
> [1] Queens are Powerful too: Mitigating Gender Bias in Dialogue Generation. EMNLP 2020.
>
> [2] Auto-Debias: Debiasing Masked Language Models with Automated Biased Prompts. ACL 2022.

---

> > ### Comment · Reviewer_Spmm · 2022-08-08
> > **Acknowledgement of Author Response**
> >
> > Thank you Authors for addressing the concerns. Overall, I think the paper is an interesting exploration of combining OOD with sparsification of pre-trained models and thus I retain my rating.

---

> ### Author Response · Authors · 2022-08-02
> **Response to Reviewer Spmm (Part 1/2)**
>
> ### Q1: No recommendation on how to predict at when to start the pruning process without actually finetuning the entire model.
> Great question! Admittedly, we do not provide a recommendation on how to predict the exact timing to start pruning, and our analysis on “the timing to start searching SRNets” mainly serves as a proof-of-concept (we will clarify it in the next version). However, we would like to stress that our finding that “we can start mask training without fully fine-tuning BERT” has its own value, which can **serve as a useful reference for the design of SRNet searching algorithm**. Moreover, it is worth noting that, to improve the efficiency of SRNet searching, we can at least directly start pruning on the pre-trained BERT, which is much more efficient than fully fine-tuning and can still produce SRNets at 50% sparsity (as shown by Sec.3.4.2). These results suggest that **we can safely reduce the amount of full BERT fine-tuning based on the available computational resources, but still find SRNets at reasonable sparsity levels**.
>
> We agree that not providing a conclusion on how to predict the exact timing is a limitation of the current manuscript. We will include a relevant discussion in the next version: A practical way is to stop full BERT fine-tuning when there is no significant improvement across several consecutive evaluation steps. In our experiments, if starting mask training from f(θ_t) is comparable to starting from fully fine-tuning, then after step t the fine-tuning performance also has little room for improvement (e.g., 20K steps of full BERT fine-tuning on MNLI). However, accurately predicting the optimal timing (the timing with the least amount of fine-tuning and comparable subnetwork performance to fully fine-tuning) is indeed intractable and we invite follow-up studies to further investigate this question.
>
> ### Q2: Presenting results as actual values.
> Thanks for the suggestion. We will include the actual values in the next version.
>
> ### Q3: Additional pre-trained models. & The effect of using large models.
> Thanks for the constructive suggestion. We have conducted additional experiments using RoBERTa. Specifically, we perform mask training with PoE loss on the standard fine-tuned RoBERTa-base. The results are shown in the following table. We can see that (1) We can find subnetworks at 50% sparsity, which outperform the full RoBERTa (std) by 6.84 points on HANS, with a relatively small drop of 1.74 on MNLI, validating that SRNets can also be found in RoBERTa. (2) Gradul sparsity increase (the last row in the following table) is also effective for RoBERTa. More detailed discussions are included in Appendix C.5 of the Rebuttal Revision. Since BERT-large is slow to train, we are still running these experiments and the results will be included in the next version.
>
> |  roberta-base   | MNLI  | HANS |
> |  ----  | ----  | ---- |
> | full model (std)     | 87.14(0.21) | 68.33(0.88) |
> | full model (poe)    | 86.56(0.18) | 76.15(1.35) |
> | mask train (poe) 0.5 | 85.40(0.14) | 75.17(0.55) |
> | mask train (poe) 0.7 | 83.48(0.29) | 68.63(1.33) |
> | mask train (poe) 0.5~0.7 | 84.41(0.15) | 71.95(1.23) |
>
> ### Q4: Results of using example reweighting and confidence regularization. & Why not consider Debiasing Focal Loss?
> Actually, we have presented the results in Appendix C.1. However, we forgot to mention them in the paper. This will be addressed in the next version.
>
> Debiasing Focal Loss is similar to example reweighting in that they both leverage the bias model’s predictions to reduce the relative importance of the most biased examples. Therefore, we do not consider Debiasing Focal Loss in our experiments.
>
> ### Q5: What does "Second, we ameliorate the mask training method ..." mean?
> It means “we refine the original mask training process”, which basically has the same meaning as our third contribution. We will rephrase it to make it more clear.

---

> > ### Author Response · Authors · 2022-08-04
> > **Results of BERT-large (Q3 to Reviewer Spmm)**
> >
> > ### Q3: Additional pre-trained models. & The effect of using large models.
> >
> > We have obtained the results of BERT-large, which are summarized in the following table. Specifically, we conduct mask training
> > with PoE loss on the standard fine-tuned BERT-large. We can see that the conclusions are basically the same as BERT-base and RoBERTa-base: (1) We can find 50% sparse SRNets from BERT-large using the original mask training. (2) Gradual sparsity increase is also effective for BERT-large. Additionally, we find that the original mask training exhibits high variance at 70% sparsity because the training fails for some random seeds. In comparison, with gradual sparsity increase, the searched subnetworks have better performance and low variance.
> >
> > |  bert-large   | MNLI  | HANS |
> > |  ----  | ----  | ---- |
> > | full model (std)     | 86.84(0.13) | 69.44(2.39) |
> > | full model (poe)    | 86.25(0.17) | 76.27(1.55) |
> > | mask train (poe) 0.5 | 85.47(0.28) | 75.40(0.64) |
> > | mask train (poe) 0.7 | 77.54(6.10) | 60.19(7.56) |
> > | mask train (poe) 0.5~0.7 | 84.83(0.26) | 70.18(2.24) |

---

### Official Review · Reviewer_6mgh · 2022-07-11

**Rating:** 5
**Confidence:** 3
**Soundness:** 2 fair
**Presentation:** 3 good
**Contribution:** 3 good

**Summary:**

The paper studies whether sparse networks pruned with different methods (i.e., mask train or IMP), or different loss functions (i.e., standard cross-entropy loss or product-of-export loss) can perform as well as unpruned models on the ID and OOD settings. The paper is presented in a very straightforward manner -- applying existing methods to various tasks in three evaluation settings and demonstrating the results for each setting. The experiments are solid and provide interesting results. The main concern with this paper is that it lacks technical contributions and deeper insights.

**Questions:**

- Why do you want to study these two problems together? What’s the motivation behind?


**Limitations:**

The paper only mentions the limitations of only conducting experiments on BERT and NLU datasets. See weaknesses above about how the paper can be further improved.


**Strengths And Weaknesses:**

Strengths:
- The paper is well organized and easy to follow in general.
- The paper studies two important problems -- seeding for subnetworks and mitigating models’ biases. Both problems are very crucial in the field and addressing these problems can have high impacts.
- The experiments are solid.

Weaknesses:
- The paper can be better motivated and better positioned. The paper claims the paper “presents the first systematic study on sparsity and dataset bias robustness for PLMs”. However, these two problems are not connected well. Each of these two problems can be viewed as an independent and important problem but I don’t see a clear motivation in the paper about why we want to study these two problems jointly.
- The paper lacks technical contributions. The paper applies existing several approaches in different evaluation settings and does not propose new methodology.
- Although the paper conducts extensive experiments, I still feel the paper lacks deeper discussion which may provide insightful ideas to facilitate further research.
- For some conclusions in the paper regarding the experimental results, it may be hasty to draw. For example, the paper suggests that the ID performance of BERT depends on memorizing the dataset bias. This is not convincing to me only based on the ID performance and the OOD performance.

=========================

I have read the responses from the authors and my concerns are partially addressed. I have reassessed this paper and updated the score.

---

> ### Author Response · Authors · 2022-08-02
> **Response to Reviewer 6mgh**
>
> ### Q1: The motivation of studying sparsity and robustness to dataset bias jointly.
> Thanks for raising the question. In the next version, we will revise relevant sections to make the motivation of our topic more clear.
>
> Jointly studying sparsity and robustness to dataset bias has a very practical reason: To achieve real-world deployment of state-of-the-art deep learning systems, the problems of robustness and efficiency should be addressed simultaneously, especially for resource-constraint and security-sensitive scenarios, e.g., self-driving cars with language interface. Therefore, this topic is “quite relevant from a usability standpoint” (by Reviewer Spmm).
>
> Most existing works study these two problems independently. For model compression, most previous studies focus on the performance drop in terms of standard ID test set accuracy, while ignoring the impact of compression on OOD generalization. For the problem of dataset bias, some studies [1] have shown that larger PLMs, which are more computationally expensive, are more robust.
>
> The above reasons motivate us, along with some recent works, to jointly study sparsity and robustness for PLMs on NLU tasks.
>
> [1] An empirical study on robustness to spurious correlations using pre-trained language models. TACL 2020.
>
> ### Q2: Lack of technical contribution and deeper insights to facilitate further research.
> We disagree on this point of weakness. **First, we would like to emphasize that our main contribution is providing novel findings for the important but underexplored area of sparsity and OOD robustness, rather than in the methodological part**. We empirically demonstrate the existence of SRNets for BERT under different training and pruning setups, which “does present an interesting direction of work” (by Reviewer Spmm) and can facilitate further research on the intersection of OOD robustness and sparsity. We would also like to mention that, in the literature, studies focusing on empirical analysis play an important role in the development of corresponding fields (e.g., the lottery ticket hypothesis [2]). Therefore, not focusing on technical innovation should not be considered a weakness and we hope that the real value of our work could be better recognized.
>
> **Second, despite building on existing methods, we do have technical contributions**: (1) We show that combining debiasing and pruning methods (especially mask training) can effectively identify SRNets in BERT, which “is quite useful from a practical standpoint for deploying models in the wild” (by Reviewer Spmm). (2) We show that gradually increasing the sparsity in mask training can find better subnetworks at higher sparsity levels. (3) We show that we can start mask training from an intermediate point of full model fine-tuning, without sacrificing the performance. Overall, our approach can serve as a useful reference and/or a strong baseline for future work on relevant topics.
>
> [2] The lottery ticket hypothesis: Finding sparse, trainable neural networks. ICLR 2019.
>
> ### Q3: The conclusion that “the ID performance of BERT depends on memorizing the dataset bias” is not convincing.
> Since the OOD dataset are specially designed by eliminating the bias (or shortcut correlation) in the training set, we believe that a large gap between the ID and OOD performance is a strong indicator that BERT depends, to a large extent, on the dataset bias to achieve good ID performance. This conclusion is also widely recognized by previous studies on dataset bias [3-5].
>
> [3] Don’t Take the Easy Way Out: Ensemble Based Methods for Avoiding Known Dataset Biases. EMNLP 2019.
>
> [4] An Empirical Study on Robustness to Spurious Correlations using Pre-trained Language Models. TACL 2020.
>
> [5] Right for the wrong reasons: Diagnosing syntactic heuristics in natural language inference. ACL 2019.

---

> > ### Comment · Reviewer_6mgh · 2022-08-09
> > **Response to the authors**
> >
> > Thanks for your detailed responses! I'm convinced that studying these two problems jointly can be useful and important. However, in my opinion, the contriubtions of this paper still do not guarantee it to be accepted to a top-tier conference. I have reassessed the paper and updated my score.

---

> ### Author Response · Authors · 2022-08-07
> **Response to Reviewer 6mgh**
>
> Hi, Reviewer 6mgh, did our response address your concerns on the motivation and technical contribution? If you have any other questions, please feel free to let us know.

---

> ### Comment · Area_Chair_d4EU · 2022-08-07
> **Please clarify weakness**
>
> Dear reviewer,
> Thank you for your review.
>
> I notice that one of the weaknesses you list is the lack of deep discussion and insightful ideas. Can you try to be more specific? Are there specific parts that you can point to where a deeper discussion would be appropriate? Any concrete questions that you would like to be discussed further? This would help the reviewer improve their work.
>
> In addition, the authors have responded to your review. Please go over their review and see if they have answered your concerns or there are remaining issues.
>
> Your AC

---

### Official Review · Reviewer_Bf72 · 2022-07-11

**Rating:** 5
**Confidence:** 4
**Soundness:** 3 good
**Presentation:** 2 fair
**Contribution:** 2 fair

**Summary:**

In this paper, the authors studied the network pruning through (1) iterative magnitude-based pruning and (2) mask training, while combing the debasing loss during the pruning process. For different settings, the paper provides thorough experiment with analysis on the results. Experiments reveal that there are subnetworks with a certain sparsity level and robustness on ODD datasets.

**Questions:**

Please check the weakness section.

**Limitations:**

I don't have concern on negative societal impact for this paper.

**Strengths And Weaknesses:**

Strength:

1. The evaluation is thorough and clear, the authors present figures where readers can see how the sparsity level is correlated with the model performance
2. The model scores reported in the figures are at a reasonably high level

Weakness:

1. Overall the results seems to be what I have expected. Take section 3.2.2 as example, (1) Existing papers have shown that the pruning can be done at around 70% sparsity level with modest performance drop. (2,3) We can consider both the imp and mask training as learnable sparsity inducing methods. We are expecting the OOD performance to be improved over standard loss by training the parameters in such a way during pruning. (4) This statement is describing the observation of experiments in this paper. How can we prove this 20% - 70% range can be generalized?

2. The conclusion that the sparse and robust BERT subnetworks exist seems to be made from experiment results on three dataset settings. In figure 2, the authors show the ID and OOD performance of pruned model using different approaches. I agree that 65% ~ 70% accuracy on HANS dataset can be a reasonably good performance.

However, is the experimental scores sufficient to draw conclusion that we have extracted subnetworks that are robust to OOD?  How do we define the robustness here? What network is robust and what is not robust?

Are we confident that the conclusion that we obtained from these datasets are generalizable?

---

> ### Author Response · Authors · 2022-08-02
> **Response to Reviewer Bf72**
>
> ### Q1: How do we define the robustness? Are the conclusions (e.g., we have extracted subnetworks that are robust to OOD) that we obtained from these datasets generalizable?
> In this paper, we focus on robustness to dataset bias for NLU tasks, and the type of bias is already known before model training. Following existing works [1-4], we consider a model as robust, if it can obviously outperform the baseline (e.g., full model trained with CE loss) in the OOD datasets and maintain comparable ID performance.
>
> To study the above topic, three OOD datasets, i.e., HANS, PAWS and Fever-Symmetric, are constructed targeting different tasks and biases and they are widely used as the benchmarks by existing works [1-4]. We also adopt the three representative OOD datasets in this work. Therefore, we are fairly confident that our main conclusions (e.g., the existence of SRNets) can be generalized to other scenarios within the same scope of our topic (e.g., for NLU datasets that also have known shortcuts/biases, using the same pruning and debiasing methods). Please refer to Q1 to Reviewer fVow for a discussion on how our method can generalize to other tasks.
>
> We agree that the 20%~70% sparsity range may not be generalizable. However, slight difference between datasets in terms of such specific observations is understandable and our focus is the existence of SRNets.
>
> [1] Don’t Take the EasyWay Out: Ensemble Based Methods for Avoiding Known Dataset Biases. EMNLP 2019.
>
> [2] Mind the trade-off: Debiasing NLU models without degrading the in-distribution performance.
>
> [3] Learning from others' mistakes: Avoiding dataset biases without modeling them. ICLR 2021.
>
> [4] Towards Debiasing NLU Models from Unknown Biases. EMNLP 2020.

---

### Official Review · Reviewer_fVow · 2022-07-17

**Rating:** 4
**Confidence:** 3
**Soundness:** 3 good
**Presentation:** 2 fair
**Contribution:** 2 fair

**Summary:**

This paper basically answers the question if there exist sparse subnetworks of pre-trained LMs that can work well under the out-of-domain setting. They focus on the BERT model and three NLU tasks and demonstrate that there exists subnetwork that do well on OOD setting for these three tasks. The findings are interesting but I found it lacks a conclusive result of how to efficiently find such sparse subnetworks.

**Questions:**

Questions:
In figure 2, which are the OOD plots and which are the ID plots?


**Limitations:**

They slightly mention the limitation of not exploring other types of pre-trained LMs and other tasks.

**Strengths And Weaknesses:**

Strengths:
- They present some interesting findings about subnetworks of PLMs: (1) there exist subnetworks that can achieve great OOD performance meanwhile preserve the in-domain (ID) performance and (2) even the fine-tuned BERT model is biased, there also exist unbiased subnetwork in it.
- They tried both IMP (iterative magnitude pruning) and Mask training, and CE loss and PoE (product of experts) loss and found that with PoE, you can directly learn effective subnetworks from the biased fine-tuned BERT model.
- They show that gradually increasing sparsity could help the training.

Weaknesses:
The proposed approach to fine-tune with biased models (PoE) seems not generalizable to arbitrary tasks. For all experiments, it uses biased models that are trained using known spurious features. I wonder how such a technique could be generalized to other tasks?

One concern I have is that, after reading the paper, I am still not sure how to find such subnetworks that do well on both ID and OOD efficiently.
- How to determine the best sparsity to trade-off performance in ID and OOD?
- How you improve the subnetwork searching process. How do you determine when to find subnetworks during the fine-tuning?

---

> ### Author Response · Authors · 2022-08-02
> **Response to Reviewer fVow**
>
> ### Q1: The proposed approach uses known spurious features. How does such a technique generalize to other tasks?
> Thanks for the good question. In this work, we focus on the scenario where the type of dataset bias is already known. This scenario is widely studied in the literature of dataset bias. Like the existing debiasing methods (e.g., PoE, example reweighting and confidence regularization that are considered in this paper) that also require prior knowledge of the type of dataset bias, our approach can generalize to other tasks/datasets by re-analyzing the type of dataset bias. Although some biases are task-specific, the spurious features can be used to train different models once they are identified. Moreover, for similar tasks, the spurious feature can also be reused (e.g., both HANS and PAWS adopt the word overlapping information as spurious features).
>
> Some recent studies have explored the unknown bias scenario. Instead of training the bias model with known spurious features, they use small models [1] or models at an early point of training [2] as the bias model. Theoretically, the pruning methods we considered can also be combined with such techniques and generalize to the unknown bias scenarios.
>
> We agree that generalizing the existence of SRNnets to other tasks/datasets is an important question. We are interested to study it in future work and relevant discussion will be included in the next version.
>
> [1] Learning from others' mistakes: Avoiding dataset biases without modeling them. ICLR 2021.
>
> [2] Towards Debiasing NLU Models from Unknown Biases. EMNLP 2020.
>
> ### Q2: How to determine the best sparsity to trade-off performance in ID and OOD?
> In all our experiments, we select the best checkpoints based on the performance on the ID dev set, without using OOD information. The selection of the best sparsity should also follow this principle. In practice, we can select the highest sparsity level where there is no obvious degradation in ID performance. In our experiments, the OOD performance is also reasonably high at such sparsity levels (approximate 50%~70%).
>
> ### Q3: How do you improve the subnetwork searching process? & How to determine when to find subnetworks during the fine-tuning?
> Thanks for this constructive feedback! In Sec.4, we show that the SRNet searching process can be improved from two aspects, i.e., gradual sparsity increase and the timing to start searching SRNets.
>
> “Gradual sparsity increase” not only improves the performance of the searched subnetworks but also **dispenses with the need to extend the training process** for high-sparsity subnetworks (see the second paragraph of Sec.4.2), which **reduces the training cost (improves efficiency)**.
>
> Our analysis of “the timing to start searching SRNets” empirically demonstrates the feasibility of starting mask training without fully fine-tuning the full BERT. Admittedly, we do not reach a conclusion on how to determine the exact timing to start mask training. Nevertheless, as a proof-of-concept (which we will clarify in the next version), the above finding still has its own values: It suggests that the efficiency of the entire training and pruning process can be improved in terms of the duration of full model fine-tuning, which can **serve as a useful reference for the design of SRNet searching algorithm**. Moreover, it is worth noting that, to improve the efficiency of SRNet searching, we can at least directly start pruning on the pre-trained BERT, which is much more efficient than fully fine-tuning and can still produce SRNets at 50% sparsity (as shown by the results of Sec.3.4.2). These findings suggest that **we can safely reduce the amount of full BERT fine-tuning based on the available computational resources, but still find SRNets at reasonable sparsity levels**.
>
> We also agree that predicting the exact timing to start mask training is still an important question and we will discuss it in the next version. To this end, a practical way is to adopt the idea of early-stopping and stop fine-tuning when there is no significant improvement across several consecutive evaluation steps. In our experiments, if starting mask training from f(θ_t) is comparable to starting from fully fine-tuning, then after step t the fine-tuning performance also has little room of improvement (e.g., 20K steps of full BERT fine-tuning on MNLI). However, accurately predicting the optimal timing is indeed intractable and we invite follow-up studies to further investigate this question.
>
> ### Q4: In figure 2, which are the OOD plots and which are the ID plots?
> The dataset names are shown in the labels of the horizontal axis of each plot. For NLI task, the ID dataset is MNLI and the OOD dataset is HANS. For paraphrase identification, the ID dataset is QQP and the OOD datasets are PAWS-qqp and PAWS-wiki. For fact verification, the ID dataset is FEVER and the OOD datasets are Fever-Symmetric v1 and v2 (Symm1 and Symm2).

---

> ### Author Response · Authors · 2022-08-07
> **Response to Reviewer fVow**
>
> Hi, Reviewer fVow, did our response address your concerns? If you have other questions, please feel free to let us know.

---

### Meta-Review · Area_Chair_d4EU · 2022-08-28

**Recommendation:** Accept
**Confidence:** Less certain

**Metareview:**

This paper proposes methods to fine sub-networks in BERT that would lead to good performance out of distribution. It considers different settings of when to search for sub-networks in the pre-train/fine-tune paradigm.

This paper has received borderline reviews, three mildly positive and one mildly negative.

The strengths in this work seem to be:
* interesting findings
* thorough evaluation
* important problem
* clarity

I agree with the importance of the problem, the very comprehensive evaluation in multiple settings, and the interesting findings, both from the perspective of interpretability and of robustness.

The reviewers noted several weaknesses, which I'll discuss below. But I think the strengths outweigh the weaknesses and the paper would make interesting contributions to the community.

Weaknesses:
* generalizability to arbitrary tasks and to other models
* how to trade off iid and ood performance
* how to choose which earlier steps to start from?

There was some discussion with the authors that have led reviewers to update their reviews. Of the weaknesses, I find the choice of tasks reasonable as they are common and well studies in OOD generalization settings. I agree with the need to experiment with other models besides BERT. The authors have added experiments with RoBERTa in their revision, for one task and setting. This makes me more confident of the applicability of the approach, but I'd suggest including similar experiments with the other tasks and settings.

The question of trade-off of iid and ood performance is inherent to the field and there's no perfect solution. The authors have made reasonable effort by not using OOD data for model selection. One concern that I still have is the use of only product-of-experts for bias mitigation, which is known to be sub-par especially in terms of trade-off, compared to Confidence Regularization. Experiments with confidence regularization would have been great to add, but I also know that the Utama et al. results are sometimes difficult to replicate. Still, consider trying it out and reporting your results.

On when to start pruning: I agree this is a major limitation that should be clearly stated and discussed. The author response gave some thoughts, please include a thoughtful discussion in the next revision.

**Award:**

No

---

### Decision · Program_Chairs · 2022-09-14

Accept